# Towards Estimating Bounds on the Effect of Policies under Unobserved Confounding

**Alexis Bellot*     Silvia Chiappa**
Google DeepMind
London, UK

## Abstract

As many practical fields transition to provide personalized decisions, data is increasingly relevant to support the evaluation of candidate plans and policies (*e.g.*, guidelines for the treatment of disease, government directives, etc.). In the machine learning literature, significant efforts have been put into developing machinery to predict the effectiveness of policies efficiently. The challenge is that, in practice, the effectiveness of a candidate policy is not always identifiable, *i.e.*, not uniquely estimable from the combination of the available data and assumptions about the domain at hand (*e.g.*, encoded in a causal graph). In this paper, we develop graphical characterizations and estimation tools to bound the effect of policies given a causal graph and observational data collected in non-identifiable settings. Specifically, our contributions are two-fold: (1) we derive analytical bounds for general probabilistic and conditional policies that are tighter than existing results, (2) we develop an estimation framework to estimate bounds from finite samples, applicable in higher-dimensional spaces and continuously-valued data. We further show that the resulting estimators have favourable statistical properties such as fast convergence and robustness to model misspecification.

## 1   Introduction

Understanding how to act upon the world around us requires humans and artificial systems to thoughtfully evaluate the effect of different plans, policies, and interventions one might consider. Data and methods that facilitate this process are increasingly relevant for high-stakes decision-making, notably in medical care with the rise of precision medicine [41], but also in education [25], law enforcement [13], public policy [44], and economics [2]. The interventions that a practitioner might consider, could consist of complex policies where a variable is set to follow a conditional or stochastic relationship depending on other variables in the system. For example, policy-makers might contemplate higher taxes on processed foods and stronger campaigns on its health risks targeted to overweight individuals. A sensible question in this context could be "what is the effect of a policy that reduces the consumption of processed foods by $50\%$ in people with a body mass index above 30?". Contrary to atomic (also called hard) interventions that force a particular value, this policy suggests a *softer* intervention.

The identifiability and estimation of policies from data is a widely studied problem in the reinforcement learning [37, 39] and causal inference [2, 12, 15, 31, 32, 40] literatures. Despite the generality entailed by many common approaches in these fields, they often rely on an impractical condition: the assumption that the effectiveness of the policy is uniquely computable from the observed data and

---

*Correspondence to Alexis Bellot: `abellot@google.com`

38th Conference on Neural Information Processing Systems (NeurIPS 2024).

assumptions about the data generating process. This can be violated in real-world settings subject to unobserved confounding, and lead to the non-identifiability of the effectiveness of a given policy. This holds irrespective of the number of samples collected so that in reality multiple answers for the effect of a policy may be equally plausible. In their seminal work in the early 1990's, Manski and Robins [27, 33] showed that, nevertheless, the effect of an atomic intervention may always be bounded in a non-trivial interval (*i.e.*, probabilities strictly contained in $[0, 1]$), irrespective of unobserved confounders or the causal structure underlying the variables involved. Causal effects are therefore in general said to be *partially identifiable*, *i.e.*, one can derive bounds that shrink the range of a priori possible values for a given effect and potentially serve as a useful support for decision-making.

Starting from this insight, the problem of partial identification has been gaining growing attention in the literature. To improve upon these bounds, one common restriction on the data generating mechanism is to assume knowledge of a causal graph describing the phenomenon of interest. Several results have derived progressively tighter bounds by exploiting the independencies in observational and interventional distributions implied by the causal graph [4, 5, 14, 46, 47, 48]. For instance, Balke and Pearl [4] (and subsequent refinements, *e.g.*, [14, 48]) defined polynomial optimization programs to derive bounds that are provably optimal. Tighter analytical bounds have also been derived in selected settings, such as for the "instrumental variable" graph [47] and discrete systems with general graphs or equivalence classes [5, 46]. These results, however, are almost exclusively concerned with atomic interventions and small systems of discretely-valued variables. Despite their generality, there is still a gap towards making the partial-identification and estimation of the effectiveness of policies practical. In particular, making scalable inferences from finite samples with continuous outcomes and higher-dimensional covariates is not possible with existing tools.

This paper aims to provide a novel estimation framework to support decision-making in non-identifiable settings, overcoming some of these challenges. We introduce several new results for the partial identification and estimation of the effect of stochastic and conditional policies from a combination of observational data and assumptions on the domain, encoded in a causal graph. Our contributions may be summarized as follows. We introduce several graphical criteria to derive new analytical bounds on the effect of policies with continuous outcomes and covariates, that improve upon the non-parametric bounds of [27, 33] and [47]. Given these analytical bounds, we then construct estimators leveraging the double machine learning [10] toolkit for scalable inference, and demonstrate that estimators exhibit favourable statistical properties such as robustness to noise and fast convergence. These results are applicable to arbitrary stochastic or conditional policies that an investigator might design, and high-dimensional covariate spaces. Recent work has made progress is developing powerful estimators for identifiable causal effects [6, 8, 19, 20, 36, 49]. However, these estimators are not applicable in non-identifiable settings. To our knowledge, the proposed estimation machinery is the first result for bounding the effect of policies given a causal diagram.

**Preliminaries.** We use capital letters to denote variables ($X$), small letters for their values ($x$), bold letters for sets of variables ($\boldsymbol{X}$) and their values ($\boldsymbol{x}$), and use supp to denote their domains of definition ($x \in \text{supp}_X$). $\mathbb{1}_x(X)$ is the indicator function that equals 1 if the statement $\{X = x\}$ is true, and equal to 0 otherwise.

The framework we use to underpin the estimation of the effect of policies rests on *structural causal models* (SCMs) [31, Def. 7.1.1]. An SCM $\mathcal{M}$ is a tuple $\langle \boldsymbol{V}, \boldsymbol{U}, \mathcal{F}, P(\boldsymbol{U}) \rangle$, where $\boldsymbol{V}$ is a set of endogenous (observed) variables, $\boldsymbol{U}$ is a set of exogenous latent variables, and $\mathcal{F} = \{f_V\}_{V \in \boldsymbol{V}}$ is a set of functions such that $f_V$ determines values of $V$ taking as argument variables $\boldsymbol{Pa}_V \subseteq \boldsymbol{V}$ and $\boldsymbol{U}_V \subseteq \boldsymbol{U}$, i.e. $V \leftarrow f_V(\boldsymbol{Pa}_V, \boldsymbol{U}_V)$. Values of $\boldsymbol{U}$ are drawn from an exogenous distribution $P(\boldsymbol{u})$. We assume the model to be recursive, i.e. that there are no cyclic dependencies among the variables. Graphically, each SCM $\mathcal{M}$ is associated with a causal diagram $\mathcal{G}$ over $\boldsymbol{V}$, where $V \rightarrow W$ if $V$ appears as an argument of $f_W$ in $\mathcal{M}$, and $V \leftarrow\!\dashrightarrow\! W$ if $\boldsymbol{U}_V \cap \boldsymbol{U}_W \neq \varnothing$, *i.e.* $V$ and $W$ share an unobserved confounder. We will use graph-theoretic family abbreviations to represent graphical relations, *e.g. an, de,* etc. Moreover, for a causal diagram $\mathcal{G}$ over $\boldsymbol{V}$, the $\boldsymbol{X}$-lower-manipulation of $\mathcal{G}$ deletes all those edges that are out of variables in $\boldsymbol{X}$, and otherwise keeps $\mathcal{G}$ as it is. The resulting graph is denoted as $\mathcal{G}_{\underline{\boldsymbol{X}}}$. The $\boldsymbol{X}$-upper-manipulation of $\mathcal{G}$ deletes all those edges that are into variables in $\boldsymbol{X}$, and otherwise keeps $\mathcal{G}$ as it is. The resulting graph is denoted as $\mathcal{G}_{\overline{\boldsymbol{X}}}$. We use $\perp\!\!\!\perp_d$ to denote $d$-separation in causal diagrams [31, Def. 1.2.3].

For a sample set $\mathcal{D} := \{\boldsymbol{v}^{(i)}\}_{i=1,\ldots,n} \sim P$, we use $\mathbb{E}_{\mathcal{D}}[f(\boldsymbol{V})] := (1/n)\sum_{i=1}^{n} f(\boldsymbol{v}^{(i)})$. We use $\|f\|_P := \sqrt{\mathbb{E}_P[\{f(\boldsymbol{V})\}^2]}$. $\widehat{f} - f = o_P(r_n)$ denotes that a function $\widehat{f}$ is a consistent estimator of $f$ at a rate $r_n$, $\widehat{f} - f = O_P(r_n)$ denotes that it is bounded in probability at rate $r_n$.

## 2  Partial Identification of the Effect of Policies

A policy $\boldsymbol{\pi}$ over a subset $\boldsymbol{X} \subset \boldsymbol{V}$ is a sequence of decision rules or plans $\boldsymbol{\pi} := \{\pi_X\}_{X \in \boldsymbol{X}}$ for determining the assignment of variables $\boldsymbol{X}$. In its most general form, every $\pi_X : \text{supp}_X \times \text{supp}_{\boldsymbol{C}_X} \to [0,1]$ is a probability mapping from domain of $\boldsymbol{C}_X \subset \boldsymbol{C} \subset \boldsymbol{V}$ to the domain of $X$.

Qualitatively different types of interventions may be modelled with $\pi_X$. Specifically, a deterministic intervention setting $X := g(\boldsymbol{c}_X)$ based on the values of $\boldsymbol{C}_X$ can be encoded as $\pi_X(x \mid \boldsymbol{c}_X) := \mathbb{1}_{g(\boldsymbol{c}_X)}(x)$ while a probabilistic intervention may be written $\pi_X(x \mid \boldsymbol{c}_X) := P_{\text{induced by } \pi_X}(X = x \mid \boldsymbol{c}_x)$. The intervened model $\mathcal{M}_{\boldsymbol{\pi}}$ represents a different regime in which the assignment $\{f_X\}_{X \in \boldsymbol{X}}$ is replaced with the assignment induced by $\boldsymbol{\pi}$. For an outcome $Y \in \boldsymbol{V}$, the interventional

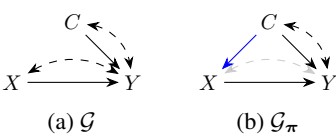

(a) $\mathcal{G}$      (b) $\mathcal{G}_{\boldsymbol{\pi}}$

Figure 1: Graphs for Example 1.

distribution $P_{\mathcal{M}_{\boldsymbol{\pi}}}(Y)$, equivalently $P_{\boldsymbol{\pi}}(Y)$, is defined as the distribution over $Y$ in $\mathcal{M}_{\boldsymbol{\pi}}$. It will be useful to adopt the notation $\bar{\boldsymbol{\pi}} := \prod_{X \in \boldsymbol{X}} \pi_X(x \mid \boldsymbol{c}_X)$ to denote the product of policy assignments on individual variables subject to intervention in $\boldsymbol{\pi}$.

**Example 1** (Illustration of policies). In the context of a public health program, let $X$ be a measure of an individual's weekly physical exercise routine, $C$ an individual's family history of metabolic diseases, and $Y$ hospital admissions related to heart disease. Consider the causal diagram Fig. 1a; currently, an individuals exercise voluntarily, which depends on unobserved factors that could be associated with risk of heart disease. The government is considering an incentive plan aimed at increasing the frequency of exercise depending individual's family history of metabolic diseases (with some probability). The proposed intervention can be encoded as $\boldsymbol{\pi} = \{\pi_X\}$ and sets the new assignment $\tilde{f}_X$ such that the variable $X$ follows the pre-specified distribution. Graphically, this policy is represented by Fig. 1b (with the new edge corresponding to the implementation of the policy highlighted in blue) and may be evaluated with the following quantity,

$$\mathbb{E}_{P_{\boldsymbol{\pi}}}[Y] = \sum_{x,c} \mathbb{E}_{P_{\boldsymbol{\pi}}}[Y \mid x, c]\pi_X(x \mid c)P_{\boldsymbol{\pi}}(c). \tag{1}$$

The inferential challenge is that this quantity is not uniquely computable from $P(x, y, c)$ and $\mathcal{G}$, requiring more complex and nuanced notions of policy evaluation.

Formally, we are interested in the evaluation of the effectiveness of a plan or policy $\boldsymbol{\pi}$, acting on a (potentially multivariate) discrete action $\boldsymbol{X}$ based on values of observed covariates $\boldsymbol{C} \in \mathbb{R}^d$, on an outcome of interest $Y \in [0, 1]^2$.

**Definition 1** (Average treatment effect). *The effectiveness of a policy $\boldsymbol{\pi}$ on $Y$ is $\mathbb{E}_{P_{\boldsymbol{\pi}}}[Y]$.*

From the investigator's perspective, only the causal diagram $\mathcal{G}$ of the environment $\mathcal{M}$ is available. No assumptions about the form or shape of $P(\boldsymbol{U})$ and $\mathcal{F}$ are made, but for the structural knowledge encoded in $\mathcal{G}$. In general, there might exist multiple SCMs $\mathcal{M}$ that induce the same causal diagram $\mathcal{G}$ and entail $P(\boldsymbol{V})$ but result in different values of $\mathbb{E}_{P_{\boldsymbol{\pi}}}[Y]$. The identification of a *set* of possible solutions that contain the true value of $\mathbb{E}_{P_{\boldsymbol{\pi}}}[Y]$ leads to the notion of *partial identification*.

**Definition 2** (Partial Identification). *The effectiveness of a policy $\boldsymbol{\pi}$ is said to be partially identifiable from $\mathcal{G}$ and $P(\boldsymbol{v})$ if*

$$\psi^{\ell}(P) \leqslant \mathbb{E}_{P_{\mathcal{M}_{\boldsymbol{\pi}}}}[Y] \leqslant \psi^u(P), \quad \text{for any } \mathcal{M} \text{ such that } \mathcal{G}_{\mathcal{M}} = \mathcal{G}, P_{\mathcal{M}}(\boldsymbol{v}) = P(\boldsymbol{v}), \tag{2}$$

*where $(\psi^{\ell}, \psi^u)$ are functionals of $P$ that are bounded away from 0 and 1, respectively.*

The earliest bound on the effect of policies, the so-called *natural bounds*, were developed considering discrete atomic interventions, i.e. of the form $\pi_X = \mathbb{1}_x(X), X \in \{1, \ldots, d_X\}$.

---

[2]An upper bound of 1 is assumed for simplicity; the following results could be generalized to more general bounded random variables.

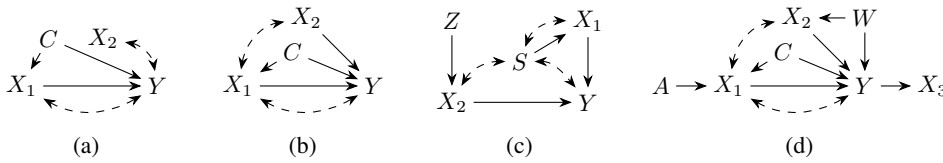

Figure 2: Graphs used in Sec. 2 and 3.

**Proposition 1** (Natural Bounds). *For any $\boldsymbol{x}$, $\mathbb{E}_P[Y \mathbb{1}_{\boldsymbol{x}}(\boldsymbol{X})] \leqslant \mathbb{E}_{P_{\boldsymbol{x}}}[Y] \leqslant \mathbb{E}_P[(Y-1)\mathbb{1}_{\boldsymbol{x}}(\boldsymbol{X})] + 1.$*

Remarkably, these inequalities hold irrespective of the causal graph of the system $\mathcal{G}$; a result that dates back to [27, 33]. Similarly, we could adapt the underlying proof strategy to derive functionals of $P$ that bound the effect of more general probabilistic or conditional policies $\boldsymbol{\pi}$, given as follows.

**Proposition 2** (Natural Policy Bounds (NPB)). *For any $\boldsymbol{\pi}$, $\mathbb{E}_P[Y\bar{\boldsymbol{\pi}}] \leqslant \mathbb{E}_{P_{\boldsymbol{\pi}}}[Y] \leqslant \mathbb{E}_P[(Y-1)\bar{\boldsymbol{\pi}}] + 1.$*

These inequalities also hold irrespective of the causal graph of the system. The natural bounds from [27, 33] are a special case of the natural policy bounds (NPBs) by setting $\boldsymbol{\pi} := \{\pi_X\}_{X \in \boldsymbol{X}}$ where $\pi_X := \mathbb{1}_x(X)$, and therefore $\bar{\boldsymbol{\pi}} = \mathbb{1}_{\boldsymbol{x}}(\boldsymbol{X})$. We could show that the NPBs are tight in some cases. For example, in Example 1, provably, no better bounds for $\mathbb{E}_{P_{\boldsymbol{\pi}}}[Y]$ than those defined by Prop. 2 could be derived (a result given in Prop. 10 in Appendix B). For other systems that involve variables that are "separated" in $\mathcal{G}$, however, better bounds may be derived by exploiting the implications of the causal graph and definition of the policy on the induced observational and interventional distributions. Consider the following example as a first illustration of this idea.

**Example 2** (Tighter bounds with causal diagram). We are interested in evaluating the effect of a policy $\boldsymbol{\pi} := \{\pi_{X_1}(x_1 \mid c), \mathbb{1}_{x_2}(X_2)\}$ from $P(x_1, c, x_2, y)$ and $\mathcal{G}$ in Fig. 2a. In particular, this problem involves a stochastic conditional intervention on $X_1$ given $C$ and a deterministic intervention on $X_2$; both variables having differing dependencies onto the rest of the system. We could show that,

$$\mathbb{E}_{P_{\boldsymbol{\pi}}}[Y] = \mathbb{E}_{P_{\pi_{X_1}}}[Y] \geqslant \mathbb{E}_P[Y\pi_{X_1}] = \psi^{\ell}(P). \tag{3}$$

The inequality gives an expression for the bound in terms of $P$ that exploits the fact that intervening on $X_2$ does not influence $Y$, and is tighter that the natural policy lower bound as $\psi^{\ell}(P) \geqslant \mathbb{E}_P[Y\pi_{X_1}]P(X_2 = x_2) = \mathbb{E}_P[Y\pi_{X_1}\mathbb{1}_{x_2}(X_2)]$ (= NPB). For the upper bound we could similarly establish that,

$$\mathbb{E}_{P_{\boldsymbol{\pi}}}[Y] \leqslant \mathbb{E}_P\left[(Y-1)\pi_{X_1}\right] + 1 = \psi^u(P). \tag{4}$$

which is smaller than the natural policy bound as $\psi^u(P) \leqslant \mathbb{E}_P[(Y-1)\pi_{X_1}\mathbb{1}_{x_2}(X_2)] + 1$ (= NPB).

This example, although relatively straightforward, serves to illustrate the potential of causal diagrams (and the constraints they imply on the effect of policies) for defining tighter bounds.

## 3   Graphical Criteria for Partial Identification

This section aims to consider more general separation statements between variables encoded in a causal diagram and the decomposition they imply to provide a systematic algorithm to bound the effectiveness of policies. We start by introducing the notion of *partial adjustment sets* (Def. 3) that is applicable with multiple intervention variables.

**Definition 3** (Partial adjustment set). *Let $\boldsymbol{\pi} := \{\pi_{\boldsymbol{X}_1}, \pi_{\boldsymbol{X}_2}\}$ be a policy on $\{\boldsymbol{X}_1, \boldsymbol{X}_2\}$ with a conditioning set $\boldsymbol{C}$. A set $\boldsymbol{W} \subseteq \boldsymbol{V} \backslash (\boldsymbol{X}_1 \cup \boldsymbol{X}_2 \cup \boldsymbol{C} \cup Y)$ is said to be a partial adjustment set for $\pi_{\boldsymbol{X}_2}$ in $\mathcal{G}$ if $(Y \perp\!\!\!\perp_d \boldsymbol{X}_2 \mid \boldsymbol{W}, \boldsymbol{C}, \boldsymbol{X}_1)_{\mathcal{G}_{\pi_{\boldsymbol{X}_1 \underline{\boldsymbol{X}_2}}}}$.*

**Proposition 3.** *Let $\boldsymbol{\pi} := \{\pi_{\boldsymbol{X}_1}, \pi_{\boldsymbol{X}_2}\}$ be a policy mapping a set of covariates $\boldsymbol{C}$ to a set of treatment variables $\{\boldsymbol{X}_1, \boldsymbol{X}_2\}$. Let $\boldsymbol{W}$ be a partial adjustment set for $\pi_{\boldsymbol{X}_2}$ in $\mathcal{G}$. Then,*

$$\mathbb{E}_P[Y\bar{\boldsymbol{\pi}}\gamma] \leqslant \mathbb{E}_{P_{\boldsymbol{\pi}}}[Y] \leqslant \mathbb{E}_P[(Y-1)\bar{\boldsymbol{\pi}}\gamma] + 1, \tag{5}$$

*where $\gamma := \gamma(\boldsymbol{X}, \boldsymbol{C}, \boldsymbol{W}) = 1/P(\boldsymbol{X}_2 \mid \boldsymbol{C}, \boldsymbol{W})$.*

In words, partial adjustment sets are designed to exploit the unconfounded status of some intervention variables with respect to the outcome, and could be leveraged to derive tighter bounds when available.

**Example 3** (Tighter bounds with partial adjustment sets)**.** Consider the problem of evaluating the effect of a policy $\boldsymbol{\pi} := \{\pi_{X_1}(x_1 \mid c), \pi_{X_2}(x_2 \mid c)\}$ from $P(x_1, c, x_2, w, y)$ and $\mathcal{G}$ in Fig. 2b. Following Def. 3, we could establish that $\boldsymbol{W} = \varnothing$ is a valid partial adjustment set for $\pi_{X_2}$ since we can verify that $(Y \perp\!\!\!\perp_d X_2 \mid C, X_2)_{\mathcal{G}_{\pi_{X_1} \underline{X_2}}}$. Prop. 3 then gives us a valid expression for bounding the effect of the policy,

$$\mathbb{E}_{P_{\boldsymbol{\pi}}}[Y] \geqslant \mathbb{E}_P\left[Y\bar{\boldsymbol{\pi}}/P(X_2 \mid C)\right] \qquad (\geqslant \mathbb{E}_P[Y\bar{\boldsymbol{\pi}}], \text{ the NPB}). \qquad (6)$$

For the upper bound we similarly find that,

$$\mathbb{E}_{P_{\boldsymbol{\pi}}}[Y] \leqslant \mathbb{E}_P\left[(Y-1)\bar{\boldsymbol{\pi}}/P(X_2 \mid C)\right] + 1 \qquad (\leqslant \mathbb{E}_P[(Y-1)\bar{\boldsymbol{\pi}}] + 1, \text{ the NPB}). \qquad (7)$$

Next, we define a second useful notion, so-called *partial conditional instrumental variables sets*, that can be exploited to evaluate bounds in $\boldsymbol{z}$-specific distributions $P(\mathbb{1}_{\boldsymbol{z}}(\boldsymbol{Z})/P(\boldsymbol{Z}))$ instead of in $P$ as described in Prop. 4.

**Definition 4** (Partial conditional instrumental set)**.** *A set $\boldsymbol{Z} \subseteq \boldsymbol{V}$ is said to be a partial instrumental set conditional on $\boldsymbol{R}$ for a policy $\boldsymbol{\pi}$ in $\mathcal{G}$ if $(Y \perp\!\!\!\perp_d \boldsymbol{Z} \mid \boldsymbol{R})_{\mathcal{G}_{\boldsymbol{\pi}}}$.*

For illustration, we give a simple criterion below to show how this subgroup structure could be exploited to derive tighter bounds.

**Proposition 4.** *Let $\boldsymbol{Z}$ be an unconditional partial instrumental set with respect to a policy $\boldsymbol{\pi}$, i.e. $(Y \perp\!\!\!\perp_d \boldsymbol{Z})_{\mathcal{G}_{\boldsymbol{\pi}}}$. Then,*

$$\max_{\boldsymbol{z}} \mathbb{E}_P[Y\bar{\boldsymbol{\pi}}\mathbb{1}_{\boldsymbol{z}}(\boldsymbol{Z})/P(\boldsymbol{Z})] \leqslant \mathbb{E}_{P_{\boldsymbol{\pi}}}[Y] \leqslant \min_{\boldsymbol{z}} \mathbb{E}_P[(Y-1)\bar{\boldsymbol{\pi}}\mathbb{1}_{\boldsymbol{z}}(\boldsymbol{Z})/P(\boldsymbol{Z})] + 1. \qquad (8)$$

The following example shows that partial adjustment sets and partial conditional instrumental sets may be usefully combined to derive tighter bounds than would be available had each proposition (Props. 3 and 4) been applied in isolation.

**Example 4** (Tighter bounds with partial adjustment and instrumental sets)**.** For this example, consider the evaluation of an atomic intervention $\boldsymbol{\pi} := \{\mathbb{1}_{x_1}(X_1), \mathbb{1}_{x_2}(X_2)\}$ in the causal diagram $\mathcal{G}$ given in Fig. 2c. Note that $\bar{\boldsymbol{\pi}} = \mathbb{1}_{\boldsymbol{x}}(\boldsymbol{X})$. Following Def. 3, $\{\varnothing\}$ is a valid partial adjustment set for $\pi_{X_1}$ since we can verify that $(Y \perp\!\!\!\perp_d X_2 \mid X_1)_{\mathcal{G}_{\pi_{X_1} \underline{X_2}}}$. Further, we could verify that $\{Z\}$ is a partial instrumental set conditional on $\{X_2\}$ with respect to $\pi_{X_1}$ since $(Y \perp\!\!\!\perp_d Z \mid X_2)_{\mathcal{G}_{\pi_{X_1}}}$. These two separation statements in (manipulated versions of) $\mathcal{G}$ could be leveraged to derive a tighter bound than previously considered:

$$\max_{\boldsymbol{z}} \mathbb{E}_P[Y\bar{\boldsymbol{\pi}}\mathbb{1}_z(Z)/P(X_2, Z)] \leqslant \mathbb{E}_{P_{\boldsymbol{\pi}}}[Y]$$
$$\leqslant \min_{\boldsymbol{z}} \mathbb{E}_P[(Y-1)\bar{\boldsymbol{\pi}}\mathbb{1}_z(Z)/P(X_2, Z)] + 1.$$

To derive bounds in a more systematic fashion, combining the notions developed so far, we present Alg. 1 that recursively seeks to find partial adjustment and partial instrumental sets in an efficient and automatic manner.

Intuitively, Alg. 1 seeks to recursively simplify the query. First by omitting the intervened variables that have no effect on the outcome; second by finding the set of intervened variables for which a partial adjustment set could be used to

---

**Algorithm 1** Bounds for the effect of policies

**Input:** Graph $\mathcal{G}$, policy $\boldsymbol{\pi}$, outcome $Y$.
**Output:** Bounds for $\mathbb{E}_{P_{\boldsymbol{\pi}}}[Y]$.

1: Let $\boldsymbol{C} = \cup_{X \in \boldsymbol{X}} \boldsymbol{C}_X, \boldsymbol{K} = \boldsymbol{V} \backslash (\boldsymbol{X} \cup \boldsymbol{C} \cup Y)$.

    /* 1. Omit redundant intervention variables. */
2: Let $\boldsymbol{R} = \boldsymbol{X} \cap An(Y)$ in $\mathcal{G}_{\boldsymbol{\pi}}$.
3: Let $\pi_{\boldsymbol{R}} := \{\pi_X(\boldsymbol{c}_X)\}_{X \in \boldsymbol{R}}$.

    /* 2. Find partial adjustment sets $\boldsymbol{W}$. */
4: Initialize $\boldsymbol{W} = \varnothing, \boldsymbol{S} = \boldsymbol{R}$.
5: **for** $R \in \boldsymbol{R}$ **do**
6:     Let $\boldsymbol{S} = \boldsymbol{S} \backslash R$.
7:     Let $\boldsymbol{W}_R = An(\boldsymbol{C} \cup Y) \cap \boldsymbol{K}$ in $\mathcal{G}_{\overline{\boldsymbol{S}}, \underline{R}}$.
8:     **if** $(Y \perp\!\!\!\perp_d R \mid \boldsymbol{C}, \boldsymbol{S}, \boldsymbol{W}, \boldsymbol{W}_R)$ in $\mathcal{G}_{\overline{\boldsymbol{S}}, \underline{R}}$ **then**
9:         $\boldsymbol{W} = \boldsymbol{W} \cup \boldsymbol{W}_R$.
10:     **end if**
11: **end for**

    /* 3. Find partial instrumental sets $\boldsymbol{Z}$. */
12: Let $\boldsymbol{T} = \{R : \boldsymbol{W}_R \in \boldsymbol{W}\}, \boldsymbol{U} = \boldsymbol{R} \backslash \boldsymbol{T}$.
13: Initialize $\boldsymbol{Z} = \varnothing$.
14: **for** $Z \in \boldsymbol{K}$ **do**
15:     **if** $(Y \perp\!\!\!\perp_d Z \mid \boldsymbol{W} \backslash Z, \boldsymbol{Z}, \boldsymbol{T}, \boldsymbol{C})$ in $\mathcal{G}_{\overline{\boldsymbol{U}}}$ **then**
16:         $\boldsymbol{Z} = \boldsymbol{Z} \cup Z$.
17:     **end if**
18: **end for**

    /* 4. Return bounds. */
19: Let $\gamma := \bar{\boldsymbol{\pi}}_{\boldsymbol{R}}\mathbb{1}_{\boldsymbol{z}}(\boldsymbol{Z})/P(\boldsymbol{T}, \boldsymbol{Z} \mid \boldsymbol{W} \backslash \boldsymbol{Z}, \boldsymbol{C})$.
20: Let $\psi_{\boldsymbol{z}}^{\ell} := \mathbb{E}_P[Y\gamma]$.
21: Let $\psi_{\boldsymbol{z}}^u := \mathbb{E}_P[(Y-1)\gamma] + 1$.
22: Return bounds: $(\max_{\boldsymbol{z}} \psi_{\boldsymbol{z}}^{\ell}, \min_{\boldsymbol{z}} \psi_{\boldsymbol{z}}^u)$.

---

tighten the bound, and third by finding the set of variables that act as valid partial instrumental sets to evaluate bounds on the most favorable conditional distributions rather than on the joint distribution.

**Proposition 5.** *Alg. 1 is sound.*

In words, Prop. 5 says that for a given causal diagram, policy $\boldsymbol{\pi}$, and joint distribution $P(\boldsymbol{v})$, the average treatment effect $\mathbb{E}_{P_{\boldsymbol{\pi}}}[Y]$ is contained in the bounds produced by Alg. 1.

**Proposition 6.** *Let $k$ be the number of variables and $m$ be the number of edges in $\mathcal{G}$. The run time of Alg. 1 is $\mathcal{O}(k(k^2 + m))$.*

**Example 5** (Steps of Alg. 1). For this example we consider evaluating a policy $\boldsymbol{\pi} := \{\pi_{X_1}(\cdot \mid C), \pi_{X_2}(\cdot \mid C), \pi_{X_3}(\cdot \mid C)\}$ from observational data $P(x_1, c, x_2, x_3, a, w, y)$ compatible with $\mathcal{G}$ in Fig. 2d by explicitly following Alg. 1. On line 1, we define $\boldsymbol{C} = \{C\}, \boldsymbol{K} = \{A, W\}$. We start by omitting potentially irrelevant intervention variables by evaluating $\boldsymbol{R} = \{X_1, X_2\}$ and noticing that $\mathbb{E}_{P_{\boldsymbol{\pi}}}[Y] = \mathbb{E}_{P_{\boldsymbol{\pi}_{\boldsymbol{R}}}}[Y]$. To find partial adjustment sets in line 4, we evaluate the for loop iterating over $\boldsymbol{R}$. For $R = X_1$, we find that $\boldsymbol{S} = X_2, \boldsymbol{W}_{X_1} = \{A, W\}$ but the if condition fails as $Y$ is not $d$-separated from $X_1$ given $\{X_2, C, A, W\}$. We turn onto $R = X_2$, where $\boldsymbol{S} = X_1, \boldsymbol{W}_{X_2} = \{A, W\}$. The if condition is triggered as $(Y \perp\!\!\!\perp_d X_2 \mid X_1, C, A, W)$ in $\mathcal{G}_{\boldsymbol{\pi}_{X_1}, \underline{X_2}}$ and therefore we update $\boldsymbol{W} = \boldsymbol{W}_{X_2} = \{A, W\}$ (giving a partial adjustment set). We continue with line 12, set $\boldsymbol{T} = \{X_2\}, \boldsymbol{U} = \{X_1\}$, and iterate over $\boldsymbol{K} = \{A, W\}$ in the for loop. For $Z = A$, we have that $(Y \perp\!\!\!\perp_d A \mid W, X_2)$ in $\mathcal{G}_{\boldsymbol{\pi}_{X_1}}$, the if condition is triggered and therefore we update $\boldsymbol{Z} = \{A\}$. For $Z = W$, we find that $(Y \not\perp\!\!\!\perp_d W \mid A, X_2)$ in $\mathcal{G}_{\boldsymbol{\pi}_{X_1}}$ and therefore terminate the for loop. Finally putting together the pieces, we evaluate the bounds to be,

$$\max_{\boldsymbol{z}} \mathbb{E}_P[Y\gamma] \leqslant \mathbb{E}_{P_{\boldsymbol{\pi}}}[Y] \leqslant \min_{\boldsymbol{z}} \mathbb{E}_P[(Y-1)\gamma] + 1, \quad \gamma := \bar{\pi}_{\boldsymbol{R}} \mathbb{1}_a(A)/P(X_2, A \mid W, C) \quad (9)$$

**Remark** (Multiple bound expressions). Alg. 1 is designed to make use of partial adjustment and conditional instrumental variable sets but it will return a single bounding expression. In general multiple different expressions could be derived for a given causal diagram and data distribution. For example, given $\mathcal{G}$ in Fig. 3 and a policy $\boldsymbol{\pi}$ on $\{X_1, X_2\}$, $\mathbb{E}_P[\bar{\pi}Y/P(X_2 \mid W_1)]$ and $\mathbb{E}_P[\bar{\pi}Y/P(X_2 \mid W_2)]$ give valid, but numerically different lower bounds for $\mathbb{E}_{P_{\boldsymbol{\pi}}}[Y]$. The superiority of one bound over another might depend on the

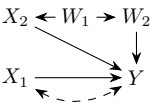

Figure 3: $\mathcal{G}$

associations in the underlying distribution $P(\boldsymbol{v})$. In this case, we could construct two different SCMs compatible with $\mathcal{G}$ that entail different orders in the numerical values for the two bounds, such that neither is always optimal. In general, we conjecture that finding an optimal analytical bound (using the proposed procedure) is undecidable from the graph alone.

The observation in this remark motivates to develop a procedure to automatically enumerate all partial adjustment sets to facilitate the search for tighter bounds. The following proposition adapts the `ListSep` algorithm by [43] to enumerate all partial adjustment sets.

**Proposition 7** (Enumerating partial adjustment sets). *Let $\boldsymbol{\pi} := \{\pi_{\boldsymbol{X}_1}, \pi_{\boldsymbol{X}_2}\}$ be a policy on $\{\boldsymbol{X}_1, \boldsymbol{X}_2\}$ with a conditioning set $\boldsymbol{C}$. All partial adjustment sets may be enumerated in time $\mathcal{O}(k(k+m))$ where $k$ are the number of variables and $m$ be the number of edges in $\mathcal{G}$.*

**Remark** (Related methods for policy evaluation). In parallel to the graphical approach for encoding structural assumptions about the domain at hand, a number of works have adopted sensitivity assumptions that quantify the degree of unobserved confounding through various data statistics, such as odds ratios, propensity scores, etc. A rich literature on sensitivity assumptions exists, including Tan's sensitivity model [38] and Rosenbaum's sensitivity model [34]. Under these models, [17, 21, 28, 45], among others, present methods that achieve validity and rate properties for the resulting estimators. These approaches start with estimators that optimize the average or conditional treatment effect bounds subject to constraints implied by the sensitivity model. Several of these works leverage Neyman orthogonality techniques to obtain rate guarantees and doubly-robustness properties. For instance, [17, 21, 28] study the estimation of bounds under Tan's sensitivity model which quantifies the degree of unobserved confounding through odds ratios, and propose estimators with various validity, sharpness, and favourable convergence rate guarantees. In contrast, [45] study the estimation of bounds under Rosenbaum's sensitivity model also deriving estimators with sharpness and fast convergence guarantees. Some works under these assumptions have also considered policy evaluation (as opposed to the evaluation of atomic interventions) under various sensitivity models in the context of Reinforcement Learning, *e.g.*, [7, 22, 23]. We interpret this line of work as complementary (applicable under different assumptions, i.e., sensitivity models rather than causal diagrams) to the techniques proposed in this paper.

## 4 Estimation of the Effect of Policies

This section aims to develop an estimation framework for the effect of a policy $\mathbb{E}_{P_\pi}[Y]$ given finite samples from $P$ that partially identifies its value. The key observation of this section is that multiple characterizations for estimation could be derived for a given bound returned by lines 20-21 in Alg. 1.

In a first instance, parameterized by the probability ratio given in Alg. 1 defined by $\boldsymbol{\gamma} = (\gamma_1, \gamma_2)$,

$$T^{\text{PW},\ell} := \mathbb{E}_P[\gamma_2 Y] \quad (= \psi_{\boldsymbol{z}}^\ell), \qquad \gamma_2 := \bar{\pi}_{\boldsymbol{U}} \gamma_1, \quad \gamma_1 := \bar{\pi}_{\boldsymbol{T}} \mathbb{1}_{\boldsymbol{z}}(\boldsymbol{Z}) / P(\boldsymbol{T}, \boldsymbol{Z} \mid \boldsymbol{W} \backslash \boldsymbol{Z}, \boldsymbol{C}). \quad (10)$$

In a second instance, parameterized by a collection of regression parameters $\boldsymbol{\mu} = (\mu_0, \mu_1, \tilde{\mu}_1, \mu_2)$ where

$$\mu_2 := \mu_2(\boldsymbol{R}, \boldsymbol{C}, \boldsymbol{W}, \boldsymbol{Z}) = \mathbb{E}_P[Y \mid \boldsymbol{R}, \boldsymbol{C}, \boldsymbol{W}, \boldsymbol{Z}], \tag{11}$$
$$\tilde{\mu}_1 := \tilde{\mu}_1(\boldsymbol{R}, \boldsymbol{C}, \boldsymbol{W}, \boldsymbol{Z}) = \bar{\pi}_{\boldsymbol{U}}(\boldsymbol{U} \mid \boldsymbol{C}) \mu_2(\boldsymbol{R}, \boldsymbol{C}, \boldsymbol{W}, \boldsymbol{Z}), \tag{12}$$
$$\mu_1 := \mu_1(\boldsymbol{T}, \boldsymbol{C}, \boldsymbol{W}, \boldsymbol{Z}) = \mathbb{E}_P[\tilde{\mu}_1(\boldsymbol{R}, \boldsymbol{C}, \boldsymbol{W}, \boldsymbol{Z}) \mid \boldsymbol{T}, \boldsymbol{C}, \boldsymbol{W}, \boldsymbol{Z}], \tag{13}$$
$$\mu_0 := \mu_0(\boldsymbol{C}, \boldsymbol{W}, \boldsymbol{Z}) = \sum_{\boldsymbol{t}} \mu_1(\boldsymbol{t}, \boldsymbol{C}, \boldsymbol{W}, \boldsymbol{Z}) \bar{\pi}_{\boldsymbol{T}}(\boldsymbol{t} \mid \boldsymbol{C}). \tag{14}$$

$T^{\text{REG},\ell} := \mathbb{E}_P[\mu_0(\boldsymbol{C}, \boldsymbol{W}, \boldsymbol{z})]$ could be shown to equal $\psi_{\boldsymbol{z}}^\ell$.

Both formulations define equivalent but different estimation targets for the lower bound[3] and may be combined leveraging the double machine learning (DML) toolkit for more efficient and robust inferences [10].

The following procedure is the main contribution of this section. It defines a DML estimator for bounding the effectiveness $\mathbb{E}_{P_\pi}[Y]$ of a policy $\boldsymbol{\pi}$.

**Definition 5** (DML Estimator). *Given $\boldsymbol{\pi}$ and $\mathcal{G}$, let $\{\boldsymbol{R}, \boldsymbol{T}, \boldsymbol{W}, \boldsymbol{C}, \boldsymbol{Z}, Y\}$ be defined as in Alg. 1. Consider a finite sample of data $\mathcal{D} \sim P$, randomly split into $K$ folds. The $k$'th partition of the sample is denoted $\mathcal{D}^{(k)}$ and $\mathcal{D}^{(-k)} := \mathcal{D} \backslash \mathcal{D}^{(k)}$. For each $k$, learn approximate nuisances $(\hat{\boldsymbol{\gamma}}_k, \hat{\boldsymbol{\mu}}_k)$ with $D^{(-k)}$. Then, define*

$$\left( \max_{\boldsymbol{z}} \ \hat{T}^{DML,\ell} \ , \ \min_{\boldsymbol{z}} \ \hat{T}^{DML,u} \right) \tag{15}$$

*to be an estimate for the bounds on $\mathbb{E}_{P_\pi}[Y]$ where,*

$$\hat{T}^{DML,\ell} := \frac{1}{K} \sum_{k=1}^{K} \mathbb{E}_{\mathcal{D}^{(k)}}[\hat{\gamma}_{2,k}\{Y - \hat{\mu}_{2,k}\}] + \mathbb{E}_{\mathcal{D}^{(k)}}[\hat{\gamma}_{1,k}\{\hat{\tilde{\mu}}_{1,k} - \hat{\mu}_{1,k}\}] + \mathbb{E}_{\mathcal{D}^{(k)}}[\hat{\mu}_{0,k}]$$

$$\hat{T}^{DML,u} := 1 + \frac{1}{K} \sum_{k=1}^{K} \mathbb{E}_{\mathcal{D}^{(k)}}[\hat{\gamma}_{2,k}\{(Y-1) - \hat{\mu}_{2,k}\}] + \mathbb{E}_{\mathcal{D}^{(k)}}[\hat{\gamma}_{1,k}\{\hat{\tilde{\mu}}_{1,k} - \hat{\mu}_{1,k}\}] + \mathbb{E}_{\mathcal{D}^{(k)}}[\hat{\mu}_{0,k}].$$

To analyse the error of the DML estimator, we consider the case that nuisances can be estimated consistently. Thus requirement is relatively mild in practice as accurate probability estimation employing off-the-shelf classification and regression methods is feasible in general. Its error with respect to the true bounds is given by the following proposition.

**Proposition 8** (Error rates). *Suppose the nuisance estimates $(\hat{\boldsymbol{\mu}}, \hat{\boldsymbol{\gamma}})$ are $L_2$-consistent and bounded. Then, the error of the DML estimator $\hat{T}^{DML} \in \{\hat{T}^{DML,\ell}, \hat{T}^{DML,u}\}$ in Def. 5 is given as follows*

$$\hat{T}^{DML} - T^{DML} = \frac{1}{K} \sum_{k=1}^{K} R_k + O_P\left(\|\hat{\gamma}_{2,k} - \gamma_2\|\|\hat{\mu}_{2,k} - \mu_2\|\right) + O_P\left(\|\hat{\gamma}_{1,k} - \gamma_1\|\|\hat{\mu}_{1,k} - \hat{\tilde{\mu}}_{1,k}\|\right)$$

*where $R_k$ is a random variable that converges to zero at a rate $\mathcal{O}_P(1/\sqrt{n})$.*

In words, the DML estimator exhibits a robustness property since the error of $\hat{T}^{\text{DML}}$ is bounded in probability at $n^{-1/2}$ rate whenever the nuisances converge at a rate $n^{-1/4}$. Note that the term

---

[3]For the upperbound, a similar construction could be derived by defining $T^{\text{PW},u} := \mathbb{E}_P[\gamma_2(Y-1)] + 1$, and $T^{\text{REG},u} := \mathbb{E}_P[\mu_0(\boldsymbol{C}, \boldsymbol{W}, \boldsymbol{z})] + 1$ with $Y$ replaced by $Y-1$ in the definition of $\mu_2$. Both estimators could be shown to be unbiased, i.e. equal to $\psi_{\boldsymbol{u}}^\ell$ defined in Alg. 1.

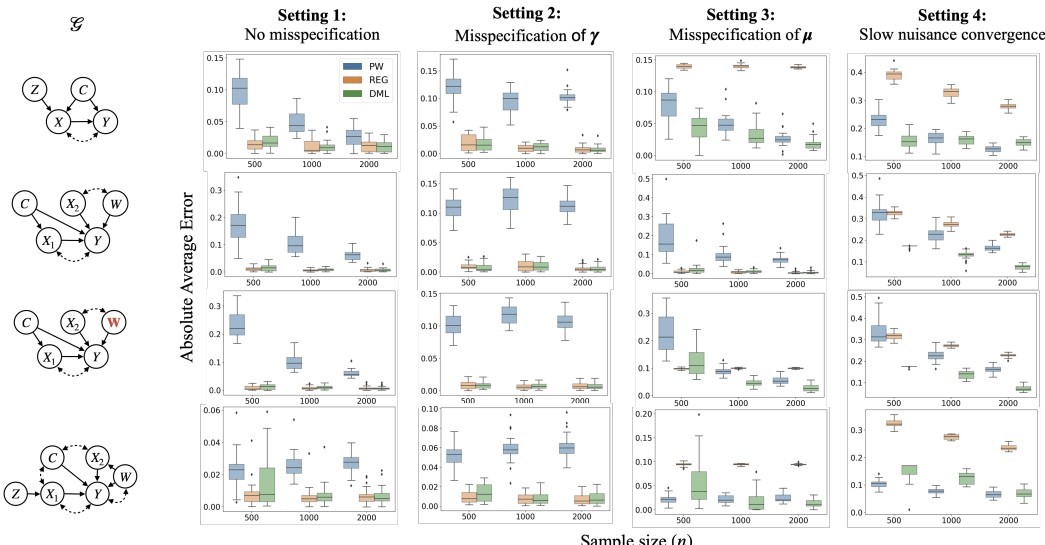

Figure 4: Experimental results on bounding the effectiveness of policies with the proposed estimators. Different rows highlight evaluations on different data generating mechanisms: the first row tests estimation with a partial instrumental set, the second row tests estimation with a partial adjustment set, the third row tests estimation with a high-dimensional partial adjustment set ($W \in \mathbb{R}^{100}$), and the fourth row tests estimation with a combination of partial adjustment and instrumental sets.

$\|\hat{\hat{\mu}}_{1,k} - \hat{\mu}_{1,k}\|$ quantifies the error in approximating the conditional expectation in Eq. (13) for a given $\hat{\hat{\mu}}_1$ estimated at that stage, and that the estimation error of $\hat{\gamma}_2$ and $\hat{\gamma}_2$ are equivalent since they are deterministic transformations of each other. The following proposition is a corollloray that shows that the DML estimator is unbiased under misspecification.

**Proposition 9** (Bias under misspecification). *Suppose either $\hat{\gamma}_1 = \gamma_1$ or $\hat{\mu}_2 = \mu_2$ and that either $\hat{\gamma}_1 = \gamma_1$ or $\hat{\hat{\mu}}_1 = \hat{\mu}_1$. Then, $\hat{T}^{DML} \in \{\hat{T}^{DML,\ell}, \hat{T}^{DML,u}\}$ is an unbiased estimator of the corresponding bound defined in Alg. 1.*

## 5 Experiments

This section evaluates the quality of policy effect estimation from finite samples. Our goal is to illustrate the computation of bounds and provide empirical evidence of the fast convergence and robustness to misspecification of estimators.

Finite sample bounds are estimated with gradient boosting classification and regression models (for conditional expectations) or by taking sample averages (for unconditional expectations). We truncated estimates of probability mass functions in the interval $[0.01, 0.99]$ to ensure positivity. We assess the quality of an estimator $\hat{T}$ by computing the absolute average error (AAE) with respect to (a proxy for) the true bounds (estimated in practice with larger sample sizes), *i.e.*, AAE $= |\hat{T}^u - \min_{\mathbf{z}} \psi_{\mathbf{z}}^u| + |\hat{T}^\ell - \max_{\mathbf{z}} \psi_{\mathbf{z}}^\ell|$. Throughout, we report various statistics: 25th, 50th, 75th percentile, etc., across evaluation runs with ten different random seeds. Further details of the simulations are provided in Appendix C.

### 5.1 Synthetic Simulations

The synthetic simulations consider 4 data generating mechanisms constructed according to the graphs in Fig. 4. They highlight the use of partial instrumental sets, partial adjustment sets, high-dimensional partial adjustment sets, and combinations of partial instrumental and adjustment sets of varying dimensionality. The task is to estimate bounds on the effectiveness of a policy $\boldsymbol{\pi} := \{\pi_{X_1}, \pi_{X_2}\}$ defined as follows.

$$\pi_{X_1} := \pi_{X_1}(X_1 = 1) = 0.5, \quad \pi_{X_2} := \pi_{X_2}(X_2 = 1 \mid c) = 1/(1 - \exp\{-c\}). \quad (16)$$

That is the policy assigns $X_1 \in \{0, 1\}$ randomly with probability 0.5, and assigns $X_2 \in \{0, 1\}$ as a function of $C$ with the probability of $X_2 = 1$ increasing with the value of $C$.

To estimate bounds on $\mathbb{E}_{P_\pi}[Y]$, we consider the proposed estimators, labelled: $T^{\text{PW}}$, estimated with the nuisances in Eq. (10), $T^{\text{REG}}$, estimated with the nuisances in Eq. (11), and $T^{\text{DML}}$ estimated with the procedure in Def. 5. Our evaluations test performance across 4 different settings designed to highlight various properties.

- **Setting 1**: All nuisances estimated correctly. This setting aims to show that all estimators converge to the bound of interest.

- **Setting 2**: Nuisances $\hat{\gamma}$ are sampled from a uniform distribution to induce misspecification in the estimation of $\gamma$.

- **Setting 3**: Nuisances $\hat{\mu}$ are sampled from a uniform distribution to induce misspecification in the estimation of $\mu$. Settings 2 and 3 aim to demonstrate the doubly-robustness property of the DML estimator.

- **Setting 4**: Noise $\epsilon$ is introduced in the estimation of all nuisances $(\gamma, \mu)$ to emphasize error due to finite sample variation. Specifically, noise $\epsilon \sim \text{Normal}(n^{-\alpha}, n^{-\alpha}), \alpha = 1/4$, that induces a slower rate of convergence as a function of sample size, inspired by [24, 20]. This setting aims to show that the fast convergence behavior of the DML estimator compared to competing estimators.

The results are given in Fig. 4[4]. We observe that across all data generating mechanisms, estimators improve with the size of the dataset and converge under no misspecification (Setting 1) to the underlying bounds. It is interesting to note also the differing accuracy of estimators in the small sample regime. $T^{\text{PW}}$, based the estimation of a ratio of probabilities, can be unstable with low sample sizes if the ratio denominator is estimated to be close to zero while $T^{\text{REG}}$, based on a sequence of regression tasks, tends to be better behaved. $T^{\text{DML}}$ in contrast is constructed as a combination of elements of $T^{\text{PW}}$ and $T^{\text{REG}}$. In particular, note in Def. 5 the use of nuisances $\mu$ and $\gamma$. As a result, $T^{\text{DML}}$ has quite a different performance profile. Settings 2 and 3 in Fig. 4 show that the DML estimator $T^{\text{DML}}$ is robust to misspecification in either the nuisances $\mu$ or $\gamma$ that highlights the robustness property. Further, when decaying noise is introduced in the estimation of nuisances (Setting 4), the DML estimator outperforms in general with a faster convergence rate. We also observe that performance remains close to optimal with high-dimensional variables $W$, demonstrating that the DML estimator provides a practical toolkit for bounding in practice.

### 5.1.1 Width of Bounds According to Different Graphical Criteria

This section evaluates the width of the bounds returned by exploiting the different graphical criteria provided in Sec. 3. The simulations are based on the data generating mechanism described by causal diagram illustrated in the fourth row of Fig. 4. We consider evaluating the policy in Eq. (16) and compute the bounds obtained by applying Prop. 2 (most conservative), Prop. 3 (using the partial adjustment set $W$ only), Prop. 4 (using the partial instrumental set $Z$ only), and finally Alg. 1 (that combines all propositions and is the proposed approach). Fig. 5 gives the results over 10 seeds of the data and across multiple data sizes, highlighting the gain achieved by exploiting the causal structure using the proposed approaches.

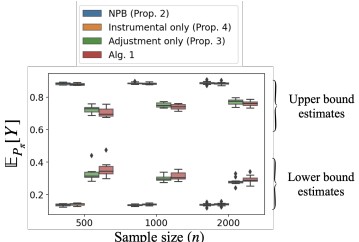

Figure 5: Width of bounds.

**Remark** (Actual width of bounds in practice). The majority of our empirical evaluations are spent on evaluating the accuracy of different methods at estimating bounds, without addressing whether the returned bounds are actually informative. In practice, the graph structure can play an important role in tightening bounds but the actual width of the bounds in a particular problem are primarily driven by the distribution of data. Here is an example to make this more concrete.

For a given policy $\pi$, Prop. 2 defines tight bounds (under some circumstances) on $\mathbb{E}_{P_\pi}[Y]$. The width of this bound is given by $1 - \mathbb{E}_P[\bar{\pi}]$ that is ultimately driven by $P(x)$. This term may therefore evaluate to anything between 0 and 1 depending on $P(x)$ and $\pi$.

---

[4]For the first row, the policy evaluated is: $\pi_X := \pi_X(X = 1 \mid c) = 1/(1 - \exp\{-c\})$.

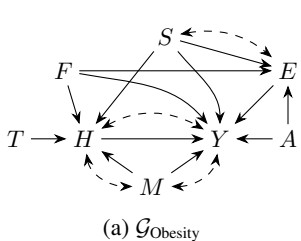

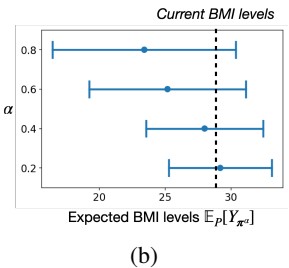

(a) $\mathcal{G}_{\mathrm{Obesity}}$

(b)

Figure 6: Health campaign evaluations.

For more complex causal structures, the bounds proposed in Props. 3 and 4 (and Alg. 1) reduce the width of the interval above. Loosely written, from $1 - \mathbb{E}_P[\bar{\pi}]$ to $1 - \mathbb{E}_P[\bar{\pi}] \times \alpha$ for some $\alpha$ that is a function of the joint distribution $P$ and the structure of the graph. But again, the actual width of the interval is ultimately determined by the values of $P$ and $\pi$ that may be large or small depending on the value probabilities involved.

## 5.2 Evaluating Health Campaigns

This section illustrates the evaluation of lifestyle recommendations for the mitigation of obesity in individuals from Colombia, Peru and Mexico [30]. The data was collected from anonymous users using a web platform, and includes reported obesity levels measured according to BMI ($Y$), age ($A$), smoking status ($S$), frequency of consumption of high caloric food ($H$), whether individuals monitored their calorie intake ($M$), family history being overweight ($F$), exercise frequency ($E$), and time using technology devices ($T$). Obesity is a multi-factored medical condition for which several causes have been acknowledged in the literature; causal diagrams relating the variables above have been curated in several related studies [1, 9]. We considered these findings to construct the causal diagram in Fig. 6a that we assumed for this example.

We aim to study the effect of a health campaign designed to lower the intake of high caloric food ($H$) and increase the frequency of exercise ($E$) on obesity levels ($Y$). For instance, we could hypothesize that the campaign leads to an increase in the observed proportion of individuals rarely consuming of high caloric food ($H$) from $0.12$ to $0.5$ and that of individuals doing exercise ($E$) regularly from $0.05$ to $0.5$. These statements could be formulated as a stochastic policy $\boldsymbol{\pi}^\alpha := \{\pi_H^\alpha, \pi_E^\alpha\}$ acting on $H$ and $E$, with new assignments,

$$\pi_H^\alpha := \pi_H^\alpha(H = \texttt{rarely}) = \alpha, \quad \pi_E^\alpha := \pi_E^\alpha(E = \texttt{regularly}) = \alpha. \tag{17}$$

We consider the evaluation of expected BMI levels $\mathbb{E}_{P_{\boldsymbol{\pi}^\alpha}}[Y]$ that range from 12 to 50 in the population, with a mean of $29.3$. First note that this or other policies acting on $(H, E)$ are not identifiable due to the bi-directed edge $\{H \leftarrow\!-\!-\!-\!\rightarrow Y\}$, but may nevertheless be bounded using Alg. 1. We find that

$$\max_t \mathbb{E}_P[Y\gamma] \leqslant \mathbb{E}_{P_{\boldsymbol{\pi}^\alpha}}[Y] \leqslant \min_t \mathbb{E}_P[(Y-1)\gamma] + 1, \quad \gamma := \bar{\pi}_{\boldsymbol{\pi}_\alpha} \mathbb{1}_t(T)/P(E, T \mid A, S, F). \tag{18}$$

To illustrate the inference of. bounds with the DML estimator, we consider evaluating policies with $\alpha = 0.2, 0.4, 0.6, 0.8$. Fig. 6b gives the results. The end-points of the intervals denote estimated lower and upper bounds. We see that policies that promote a healthier lifestyle (larger values of $\alpha$) are expected to reduce obesity levels on average but substantial uncertainty is still expected.

## 6 Conclusions

The evaluation of policies is arguably one of the critical ingredients enabling more personalized decision-making systems. When the effect of policies is not identifiable, bounds can provide an effective support for making informed decisions. In this paper we developed partial identification and estimation tools for bounding the effect of a (stochastic or conditional) policy given data and assumptions encoded in a causal graph. We introduced several graphical characterizations that induce tighter bounds, and developed an estimation framework that exhibit robustness to noise and fast convergence. The results of this paper were illustrated through synthetic simulations and a real-world health campaign example for the reduction of obesity levels.

## Acknowledgements

We thank the anonymous reviewers for helpful comments.

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

# Appendix

## Table of Contents

# A   Preliminaries, Related Work, and Impact Statement

## A.1   Preliminaries

For the derivation of results, we use the notion of counterfactuals defined as follows.

For an SCM $\mathcal{M}$, arbitrary subsets of endogenous variables $X, Y$, the potential outcome of $Y$ to intervention $do(x)$, denoted by $Y_x(u)$, is the solution for $Y$ with $U = u$ in the sub-model $\mathcal{M}_x$. It can be read as the counterfactual sentence "the value that $Y$ would have obtained in situation $U = u$, had $X$ been $x$.". Statistically, averaging $u$ over the distribution $P(u)$ leads to the counterfactual variables $Y_x$. The distribution of the variable $Y_x$ is denoted $P(Y_x)$. With this formalism, we can express all the quantities in the main body of this paper in the language of counterfactuals. For instance, the distribution of $Y$ in sub-model $\mathcal{M}_x$ can alternatively be written $P(Y_x) \equiv P_x(Y)$. Moreover, $\mathbb{E}_P[f(Y_x) \mid z]$, also written $\mathbb{E}_{P(\cdot|z)}[f(Y_x)]$, denotes the conditional expectation of $f(Y)$ over $P(y_x \mid z)$. Similarly, the distribution over $Y$ in the sub-model $\mathcal{M}_\pi$ can be written $P_\mathcal{M}(Y_\pi)$, or $P(Y_\pi) \equiv P_\pi(Y)$ for short. See [31, Chapter 7] for further context on counterfactuals.

**Definition 6** (The Axioms of Counterfactuals, Chapter 7.3.1 [31]). *For any three sets of endogenous variables $X, Y, W$ in a causal model and $x, w$ in the domains of $X$ and $W$, the following holds:*

- *Composition: $W_x = w$ implies that $Y_{x,w} = Y_x$.*

- *Effectiveness: $X_{w,x} = x$.*

- *Reversibility: $Y_{x,w} = y$ and $W_{x,y} = w$ imply that $Y_x = y$.*

**Theorem 1** (Soundness and Completeness of the Axioms Theorems 7.3.3, 7.3.6 [31]). *The Axioms of counterfactuals are sound and complete for all causal models.*

The following rules to manipulate experimental distributions produced by policies extend the do-calculus and will be used for the proof of several theoretical statements [12].

**Theorem 2** (Inference Rules $\sigma$-calculus [12]). *Let $\mathcal{G}$ be a causal diagram compatible with an SCM $\mathcal{M}$, with endogenous variables $V$. For any disjoint subsets $X, Y, Z \subseteq V$, two disjoint subsets $T, W \subseteq V \backslash (Z \cup Y)$ (i.e., possibly including $X$), the following rules are valid for any intervention strategies $\pi_X$, $\pi_Z$, and $\pi'_Z$ such that $\mathcal{G}_{\pi_X \pi_Z}$, $\mathcal{G}_{\pi_X \pi'_Z}$ have no cycles:*

- *Rule 1 (Insertion/Deletion of observations):*

$$P_{\pi_X}(y \mid w, t) = P_{\pi_X}(y \mid w) \quad \text{if } (T \perp\!\!\!\perp_d Y \mid W) \text{ in } \mathcal{G}_{\pi_X}.$$

- *Rule 2 (Change of regimes under observation):*

$$P_{\pi_X, \pi_Z}(y \mid z, w) = P_{\pi_X, \pi'_Z}(y \mid z, w) \quad \text{if } (Y \perp\!\!\!\perp_d Z \mid W) \text{ in } \mathcal{G}_{\pi_X, \pi_Z, \underline{Z}} \text{ and } \mathcal{G}_{\pi_X, \pi'_Z, \underline{Z}}$$

- *Rule 3 (Change of regimes without observation):*

$$P_{\pi_X, \pi_Z}(y \mid w) = P_{\pi_X, \pi'_Z}(y \mid w) \quad \text{if } (Y \perp\!\!\!\perp_d Z \mid W) \text{ in } \mathcal{G}_{\pi_X, \pi_Z, \overline{Z(W)}} \text{ and } \mathcal{G}_{\pi_X, \pi'_Z, \overline{Z(W)}}$$

*where $Z(W)$ is the set of elements in $Z$ that are not ancestors of $W$ in $\mathcal{G}_{\pi_X}$.*

## A.2   Related Work

Active experimentation by physically manipulating reality is generally not feasible for many consequential applications. This motivates the study of causal effect identification and estimation. The former studies the problem of inferring whether a unique expression or bound of the desired causal effect could be derived from the available data and assumptions. The latter studies the problem of providing efficient estimators from finite samples to compute causal effects or bounds on causal effects in practice. In the following, we review these two lines of work to better contextualize our contribution.

**Partial identification**. The natural bounds were derived to demonstrate that useful inference about causal effects could be drawn without making identifying assumptions beyond the observed data [27]. Relatedly, an analysis of bounds of causal effects was also provided for studies with imperfect

compliance under a set of instrumental variable assumptions that are not sufficient however to identify the causal effect of interest [33]. These bounds may be written in closed form and have recently been extended as a more general strategy for bounding causal effects given assumptions encoded in causal diagrams in discrete systems [5, 46]. Recent work has also considered bounds in closed form with access to both observational and interventional distributions [18]. These techniques were developed alongside a second line of research that employs a polynomial optimization program to compute causal bounds given a causal diagram [4]. They proposed a family of canonical models parameterized according to the causal diagram, reducing the bounding problem to a series of equivalent linear programs. [11] further used Bayesian techniques to investigate the sharpness of these bounds with regard to the observational sample size. Recently, [14, 48] describe a polynomial programming approach to solve the partial identification for general causal graphs. More recent proposals consider parameterizations in the space of linear combinations of a set of fixed basis functions [29] and neural networks [3, 16].

In parallel, a number of works have adopted sensitivity assumptions (as an alternative or in combination with a causal diagram) that quantify the degree of unobserved confounding through various data statistics, such as odds ratios, propensity scores, etc. A rich literature on sensitivity assumptions exists, including Tan's sensitivity model [38] and Rosenbaum's sensitivity model [34]. Under these models, [17, 21, 28, 45], among others, present methods that achieve validity and rate properties for the resulting estimators. These approaches start with estimators that optimize the average or conditional treatment effect bounds subject to constraints implied by the sensitivity model. Several of these works leverage Neyman orthogonality techniques to obtain rate guarantees and doubly-robustness properties. For instance, [17, 21, 28] study the estimation of bounds under Tan's sensitivity model which quantifies the degree of unobserved confounding through odds ratios, and propose estimators with various validity, sharpness, and favourable convergence rate guarantees. In contrast, [45] study the estimation of bounds under Rosenbaum's sensitivity model also deriving estimators with sharpness and fast convergence guarantees.

While some works have considered policy evaluation under various sensitivity models in the context of Reinforcement Learning, e.g., [7, 22, 23], most works target estimation of bounds on the effect of atomic interventions. We interpret this line of work as complementary (applicable under different assumptions, i.e., sensitivity models rather than causal diagrams) to the techniques proposed in this paper.

**Causal effect estimation.** With a causal diagram as input, causal effect estimation has traditionally focused on a subset of identifiable scenarios, relying on assumptions such as backdoor criterion or the availability of adjustment sets. Identification expressions in those cases are given by (sequential) covariate adjustments that have lead to statistically appealing estimators from observational data [32]. Notable examples are doubly robust estimators [10, 35, 42]. Recently these techniques have been extended to settings identifiable from multiple experimental distributions [8, 19, 20]. To our knowledge, no estimation framework specific to the estimation of analytical expressions derived for bounds on the effect of policies or atomic interventions has been developed.

### A.3 Broader Impact Statement

Our work investigates the conditions under which policies may be bounded from observational data. Issues of policy evaluation are central to fields of application involved in decision-making as well as AI and ML. We believe that a tool to bound the effect of stochastic and conditional policies in systems with unobserved confounding is an important addition to the scientific toolbox. Reasoning instead without acknowledging for the potential of unobserved confounding may lead researchers to operate on a more heuristical basis. For instance, the hypothetical effect of public policies might be misrepresented if informed by the evaluation of an idealized atomic intervention that is difficult to implement in practice. And, once implemented, might have unintended consequences. With this background, we believe that research on the partial identification and estimation of policies based on stochastic or conditional interventions can help scientists and individuals make more informed decisions.

In this work, we start from the assumption that a causal diagram that is consistent with the underlying data generating system of interest is available. In general, this requires domain knowledge. While some of the bounds provided do not require full knowledge of the causal graph, whenever a $d$-separation is assumed its truth value should be justified by prior knowledge or experiment. It is

important also to make the distinction between the task of partial identification, that is inferring an expression for bounds on causal effects, and that of estimation, that is providing efficient estimators from finite samples to compute bounds in practice. This set of results concerns both of these tasks. The first objective of our procedure is to provide an expression for lower and upper bounds, irrespective of the accuracy with which one is able to approximate $P(\boldsymbol{V})$ from finite samples. The second objective is introduce efficient estimators for bounds on the effect of policies using finite samples from $P(\boldsymbol{V})$. In higher-dimensional systems, the computational complexity of estimating the conditional distributions that define lower and upper bounds on causal effects is a substantial challenge. Consequently, practitioners must exercise caution when deploying the proposed method in small sample scenarios where estimators may be inaccurate. Moreover, we have stated our convergence guarantees in the infinite sample limit, without quantifying the finite-sample estimation uncertainty. Finite-sample properties could be explored similarly to [24] given a particular choice of function class to extend our results with high-probability bounds. Finally, we emphasize that simulations on real and synthetic data are provided for illustration purposes only. These results do not recommend or advocate for the implementation of a particular policy, and should be considered in practice in combination with other aspects of the decision-making process.

# B Proofs

This section presents proofs for theoretical statements in the main body of this paper.

## B.1 Proofs of statements in Sec. 2

**Prop. 1 restated.** *For any $\boldsymbol{x}$, $\mathbb{E}_P[Y\mathbb{1}_{\boldsymbol{x}}(\boldsymbol{X})] \leqslant \mathbb{E}_{P_{\boldsymbol{x}}}[Y] \leqslant \mathbb{E}_P[(Y-1)\mathbb{1}_{\boldsymbol{x}}(\boldsymbol{X})] + 1$.*

*Proof.* Consider the derivation of the lower bound.

$$
\begin{aligned}
\mathbb{E}_P[Y_{\boldsymbol{x}}] &= \sum_{\boldsymbol{x}'} \mathbb{E}_P[Y_{\boldsymbol{x}} \mid \boldsymbol{x}'] P(\boldsymbol{x}') \\
&\geqslant \mathbb{E}_P[Y_{\boldsymbol{x}} \mid \boldsymbol{x}] P(\boldsymbol{x}) \\
&\overset{(1)}{=} \mathbb{E}_P[Y \mid \boldsymbol{x}] P(\boldsymbol{x}) \\
&= \sum_y y P(y \mid \boldsymbol{x}) P(\boldsymbol{x}) \\
&= \sum_{y,x} y \mathbb{1}_{\boldsymbol{x}}(\boldsymbol{x}) P(y, \boldsymbol{x}) \\
&= \mathbb{E}_P[Y\mathbb{1}_{\boldsymbol{x}}(\boldsymbol{X})]
\end{aligned}
$$

(1) follows by the composition axiom of counterfactuals.

Consider the derivation of the upper bound. It holds that,

$$
\begin{aligned}
\mathbb{E}_P[Y_{\boldsymbol{x}}] &= \sum_{\boldsymbol{x}'} \mathbb{E}_P[Y_{\boldsymbol{x}} \mid \boldsymbol{x}'] P(\boldsymbol{x}') \\
&= \mathbb{E}_P[Y_{\boldsymbol{x}} \mid \boldsymbol{x}] P(\boldsymbol{x}) + \sum_{\boldsymbol{x}' \neq \boldsymbol{x}} \mathbb{E}_P[Y_{\boldsymbol{x}} \mid \boldsymbol{x}'] P(\boldsymbol{x}') \\
&\overset{(1)}{\leqslant} \mathbb{E}_P[Y \mid \boldsymbol{x}] P(\boldsymbol{x}) + \sum_{\boldsymbol{x}' \neq \boldsymbol{x}} P(\boldsymbol{x}') \\
&= \mathbb{E}_P[Y \mid \boldsymbol{x}] P(\boldsymbol{x}) + 1 - P(\boldsymbol{x}) \\
&= \mathbb{E}_P[Y\mathbb{1}_{\boldsymbol{x}}(\boldsymbol{X})] + 1 - \mathbb{E}_P[\mathbb{1}_{\boldsymbol{x}}(\boldsymbol{X})] \\
&= \mathbb{E}_P[(Y-1)\mathbb{1}_{\boldsymbol{x}}(\boldsymbol{X})] + 1.
\end{aligned}
$$

(1) follows by the boundedness of $Y$, here assumed bounded by 1 for simplifying the derivation. $\qquad\square$

**Proposition 10.** *The natural bounds are tight in general.*

*Proof.* We show this statement by providing a pair of SCMs that agree on the input observational distribution but evaluate to the lower and upperbounds, respectively, specified by the natural bounds. Let $\mathbb{M}(\mathcal{G})$ denote the space of SCMs that induce the causal diagram $\mathcal{G}$ in Fig. 1a. We introduce a pair of SCMs compatible with the causal diagram $\mathcal{G}$ that evaluate to lower and upper bounds respectively. Let $X$ be binary, $Y \in [0, 1]$, and $C \in \mathbb{R}$, and consider $\mathcal{M}_1, \mathcal{M}_2 \in \mathbb{M}(\mathcal{G})$ defined as follows,

$$
\mathcal{M}_1 := \begin{cases} x := f_X(u_1) \\ c := f_C(u_2) \\ y := \begin{cases} f_Y(x, c, u_1, u_2) & \text{if } x = f_X(u_1), \\ 0 & \text{otherwise.} \end{cases} \end{cases}
$$

and,

$$
\mathcal{M}_2 := \begin{cases} x := f_X(u_1) \\ c := f_C(u_2) \\ y := \begin{cases} f_Y(x, c, u_1, u_2) & \text{if } x = f_X(u_1), \\ 1 & \text{otherwise.} \end{cases} \end{cases}
$$

For a $P_{\mathcal{M}_1}(\boldsymbol{u}) = P_{\mathcal{M}_2}(\boldsymbol{u})$, both SCMs agree on observational distributions $P_{\mathcal{M}_1}(x, y, c) = P_{\mathcal{M}_2}(x, y, c)$. However the following derivations show that the interventional expectation $\mathbb{E}_{P_{\mathcal{M}_1}}[Y \mid do(x = 1)]$ differs across models: for $\mathcal{M}_1$ equal to the analytical lower bound, and for $\mathcal{M}_2$ equal to the analytical upper bound demonstrating that (in this case) the bound is tight. In particular,

$$
\begin{aligned}
\mathbb{E}_{P_{\mathcal{M}_1}}&[Y \mid do(x = 1)] \\
&= \mathbb{E}_{P_{\mathcal{M}_1}}[Y \mid x = 1, u : x = f_X(u)]P(u : x = f_X(u)) \\
&+ \mathbb{E}_{P_{\mathcal{M}_1}}[Y \mid x = 1, u : x \neq f_X(u)]P(u : x \neq f_X(u)) \\
&= \mathbb{E}_{P_{\mathcal{M}_1}}[Y \mid x = 1]P(x = 1) \\
&= \mathbb{E}_P[Y \mid x = 1]P(x = 1) \\
&= \mathbb{E}_P[Y \mathbb{1}_1(X)] \\
\mathbb{E}_{P_{\mathcal{M}_2}}&[Y \mid do(x = 1)] \\
&= \mathbb{E}_{P_{\mathcal{M}_2}}[Y \mid x = 1, u : x = f_X(u)]P(u : x = f_X(u)) \\
&+ \mathbb{E}_{P_{\mathcal{M}_2}}[Y \mid x = 1, u : x \neq f_X(u)]P(u : x \neq f_X(u)) \\
&= \mathbb{E}_{P_{\mathcal{M}_2}}[Y \mid x = 1]P(x = 1) + \mathbb{E}_{P_{\mathcal{M}_2}}[Y \mid x = 1, u : x = f_X(u)]P(u : x = f_X(u)) \\
&= \mathbb{E}_{P_{\mathcal{M}_2}}[Y \mid x = 1]P(x = 1) + 1 - P(x = 1) \\
&= \mathbb{E}_P[(Y - 1)\mathbb{1}_1(X)] + 1.
\end{aligned}
$$

$\square$

**Prop. 2 restated.** *For any $\boldsymbol{\pi}$, $\mathbb{E}_P[Y\bar{\boldsymbol{\pi}}] \leqslant \mathbb{E}_{P_{\boldsymbol{\pi}}}[Y] \leqslant \mathbb{E}_P[(Y - 1)\bar{\boldsymbol{\pi}}] + 1$.*

*Proof.* Consider the derivation of the lower bound. Let $\boldsymbol{\pi}$ denote a policy from an arbitrary set of covariates $\boldsymbol{C}$ to $\boldsymbol{X}$. By marginalizing over $\boldsymbol{X}, \boldsymbol{C}$ we find that

$$
\mathbb{E}_{P_{\boldsymbol{\pi}}}[Y] = \mathbb{E}_P[Y_{\boldsymbol{\pi}}] = \int \mathbb{E}_P[Y_{\boldsymbol{\pi}} \mid \boldsymbol{x}, \boldsymbol{c}] \prod_{X \in \boldsymbol{X}} \pi_X(x \mid \boldsymbol{c}_X)P(\boldsymbol{c}) \, d\boldsymbol{c}d\boldsymbol{x}.
$$

$C_X$ denotes the subset of $\boldsymbol{C}$ that is used to inform the intervention on $X$. Moreover,

$$
\begin{aligned}
\mathbb{E}_P[Y_{\boldsymbol{\pi}} \mid \boldsymbol{x}, \boldsymbol{c}] &\overset{(1)}{=} \mathbb{E}_P[Y_{\boldsymbol{x}} \mid \boldsymbol{c}] \\
&= \sum_{\boldsymbol{x}'} \mathbb{E}_P[Y_{\boldsymbol{x}} \mid \boldsymbol{x}', \boldsymbol{c}]P(\boldsymbol{x}' \mid \boldsymbol{c}) \\
&\geqslant \mathbb{E}_P[Y_{\boldsymbol{x}} \mid \boldsymbol{x}, \boldsymbol{c}]P(\boldsymbol{x} \mid \boldsymbol{c}) \\
&\overset{(2)}{=} \mathbb{E}_P[Y \mid \boldsymbol{x}, \boldsymbol{c}]P(\boldsymbol{x} \mid \boldsymbol{c}).
\end{aligned}
$$

(1) holds by Rule 2 in Thm. 2 by swapping the policy $\boldsymbol{\pi}$ by the do intervention $do(\boldsymbol{X} = \boldsymbol{x})$: given that the policy acts on $\boldsymbol{X}$ taking as inputs $\boldsymbol{C}$ it is always true that $Y \perp\!\!\!\perp_d \boldsymbol{X} \mid \boldsymbol{C}$ in $\mathcal{G}_{\boldsymbol{\pi}\underline{\boldsymbol{X}}}$ and $\mathcal{G}_{\underline{\boldsymbol{X}}\overline{\boldsymbol{X}}}$ in any graph $\mathcal{G}$. The following equality follows by marginalizing over the domain of $\boldsymbol{X}$, which is assumed discrete. (2) follows by the composition axiom of counterfactuals. By combining these two expressions we get

$$
\begin{aligned}
\mathbb{E}_P[Y_{\boldsymbol{\pi}}] &\geqslant \int \prod_{X \in \boldsymbol{X}} \pi(x \mid \boldsymbol{c}_X)P(\boldsymbol{c})\mathbb{E}_P[Y \mid \boldsymbol{x}, \boldsymbol{c}]P(\boldsymbol{x} \mid \boldsymbol{c}) \, d\boldsymbol{c}d\boldsymbol{x} \\
&= \int \bar{\boldsymbol{\pi}}yP(y \mid \boldsymbol{x}, \boldsymbol{c})P(\boldsymbol{x} \mid \boldsymbol{c})P(\boldsymbol{c}) \, dyd\boldsymbol{c}d\boldsymbol{x} \\
&= \mathbb{E}_P[Y\bar{\boldsymbol{\pi}}].
\end{aligned}
$$

Consider the derivation of the upper bound. It holds that,

$$\mathbb{E}_P[Y_{\boldsymbol{\pi}} \mid \boldsymbol{x}, \boldsymbol{c}] = \mathbb{E}_P[Y_{\boldsymbol{x}} \mid \boldsymbol{c}]$$

$$= \sum_{\boldsymbol{x}'} \mathbb{E}_P[Y_{\boldsymbol{x}} \mid \boldsymbol{x}', \boldsymbol{c}]P(\boldsymbol{x}' \mid \boldsymbol{c})$$

$$= \mathbb{E}_P[Y_{\boldsymbol{x}} \mid \boldsymbol{x}, \boldsymbol{c}]P(\boldsymbol{x} \mid \boldsymbol{c}) + \sum_{\boldsymbol{x}' \neq \boldsymbol{x}} \mathbb{E}_P[Y_{\boldsymbol{x}} \mid \boldsymbol{x}', \boldsymbol{c}]P(\boldsymbol{x}' \mid \boldsymbol{c})$$

$$\overset{(1)}{\leqslant} \mathbb{E}_P[Y \mid \boldsymbol{x}, \boldsymbol{c}]P(\boldsymbol{x} \mid \boldsymbol{c}) + \sum_{\boldsymbol{x}' \neq \boldsymbol{x}} P(\boldsymbol{x}' \mid \boldsymbol{c})$$

$$= \mathbb{E}_P[Y \mid \boldsymbol{x}, \boldsymbol{c}]P(\boldsymbol{x} \mid \boldsymbol{c}) + 1 - P(\boldsymbol{x} \mid \boldsymbol{c}).$$

(1) follows by the boundedness of $Y$, here assumed bounded by 1 for simplifying the derivation. As above by combining this inequality with the decomposition of the policy effect we get

$$\mathbb{E}_{P_{\boldsymbol{\pi}}}[Y] \leqslant \int \prod_{X \in \boldsymbol{X}} \pi(x \mid \boldsymbol{c}_X)P(\boldsymbol{c}) \left\{ \mathbb{E}_P[Y \mid \boldsymbol{x}, \boldsymbol{c}]P(\boldsymbol{x} \mid \boldsymbol{c}) + 1 - P(\boldsymbol{x} \mid \boldsymbol{c}) \right\} \, d\boldsymbol{c} d\boldsymbol{x}$$

$$= \mathbb{E}_P[Y\bar{\boldsymbol{\pi}}] + 1 - \mathbb{E}_P[\bar{\boldsymbol{\pi}}]$$

$$= \mathbb{E}_P[(Y - 1)\bar{\boldsymbol{\pi}}] + 1.$$

$\square$

**Prop. 3 restated**. *Let $\boldsymbol{\pi} := \{\pi_{\boldsymbol{X}_1}, \pi_{\boldsymbol{X}_2}\}$ be a policy mapping a set of covariates $\boldsymbol{C}$ to a set of treatment variables $\{\boldsymbol{X}_1, \boldsymbol{X}_2\}$. Let $\boldsymbol{W}$ be a partial adjustment set for $\pi_{\boldsymbol{X}_2}$ in $\mathcal{G}$. Then,*

$$\mathbb{E}_P[Y\bar{\boldsymbol{\pi}}\gamma] \leqslant \mathbb{E}_{P_{\boldsymbol{\pi}}}[Y] \leqslant \mathbb{E}_P[(Y - 1)\bar{\boldsymbol{\pi}}\gamma] + 1,$$

*where $\gamma := \gamma(\boldsymbol{X}, \boldsymbol{C}, \boldsymbol{W}) = 1/P(\boldsymbol{X}_2 \mid \boldsymbol{C}, \boldsymbol{W})$.*

*Proof.* Consider the derivation of the lower bound. By marginalizing over $\boldsymbol{X}, \boldsymbol{C}$ we find that

$$\mathbb{E}_{P_{\boldsymbol{\pi}}}[Y] = \mathbb{E}_P[Y_{\boldsymbol{\pi}}] = \int \mathbb{E}_P[Y_{\boldsymbol{\pi}} \mid \boldsymbol{x}, \boldsymbol{c}] \prod_{X \in \boldsymbol{X}} \pi_X(x \mid \boldsymbol{c}_X)P(\boldsymbol{c}) \, d\boldsymbol{c} d\boldsymbol{x}.$$

$\boldsymbol{C}_X$ denotes the subset of $\boldsymbol{C}$ that is used to inform the intervention on $X$. Denote $\bar{\boldsymbol{\pi}} := \prod_{X \in \boldsymbol{X}} \pi_X(x \mid \boldsymbol{c}_X)$. Let $\boldsymbol{X}_1, \boldsymbol{X}_2$ be a partition of $\boldsymbol{X}$ such that $\boldsymbol{W} \subseteq \boldsymbol{V} \backslash (\boldsymbol{X} \cup \boldsymbol{C} \cup Y)$ is partial adjustment set for $\pi_{\boldsymbol{X}_2}$. Then,

$$\mathbb{E}_P[Y_{\boldsymbol{\pi}} \mid \boldsymbol{x}, \boldsymbol{c}] = \mathbb{E}_P[Y_{\pi_{\boldsymbol{X}_1}, \pi_{\boldsymbol{X}_2}} \mid \boldsymbol{x}, \boldsymbol{c}]$$

$$= \int \mathbb{E}_P[Y_{\boldsymbol{x}_1, \boldsymbol{x}_2} \mid \boldsymbol{c}, \boldsymbol{w}]P(\boldsymbol{w} \mid \boldsymbol{c}) \, d\boldsymbol{w}$$

$$\overset{(1)}{=} \int \mathbb{E}_P[Y_{\boldsymbol{x}_1} \mid \boldsymbol{x}_2, \boldsymbol{w}, \boldsymbol{c}]P(\boldsymbol{w} \mid \boldsymbol{c}) \, d\boldsymbol{w}$$

$$= \int \sum_{\boldsymbol{x}_1'} \left\{ \mathbb{E}_P[Y_{\boldsymbol{x}_1} \mid \boldsymbol{x}_1', \boldsymbol{x}_2, \boldsymbol{w}, \boldsymbol{c}]P(\boldsymbol{x}_1' \mid \boldsymbol{x}_2, \boldsymbol{w}, \boldsymbol{c}) \right\} P(\boldsymbol{w} \mid \boldsymbol{c}) \, d\boldsymbol{w}$$

$$\geqslant \int \mathbb{E}_P[Y_{\boldsymbol{x}_1} \mid \boldsymbol{x}_1, \boldsymbol{x}_2, \boldsymbol{w}, \boldsymbol{c}]P(\boldsymbol{x}_1 \mid \boldsymbol{x}_2, \boldsymbol{w}, \boldsymbol{c})P(\boldsymbol{w} \mid \boldsymbol{c}) \, d\boldsymbol{w}$$

$$= \int \mathbb{E}_P[Y \mid \boldsymbol{x}_1, \boldsymbol{x}_2, \boldsymbol{w}, \boldsymbol{c}]P(\boldsymbol{x}_1 \mid \boldsymbol{x}_2, \boldsymbol{w}, \boldsymbol{c})P(\boldsymbol{w} \mid \boldsymbol{c}) \, d\boldsymbol{w}.$$

(1) follows by the definition of $\boldsymbol{X}_2$ as a partial adjustment set: it holds that $(Y \perp\!\!\!\perp_d \boldsymbol{X}_2 \mid \boldsymbol{W}, \boldsymbol{C}, \boldsymbol{X}_1)$ in $\mathcal{G}_{\overline{\boldsymbol{X}_1 \boldsymbol{X}_2}}$ which induces the equality $\mathbb{E}_P[Y_{\boldsymbol{x}_1, \boldsymbol{x}_2} \mid \boldsymbol{c}, \boldsymbol{w}] = \mathbb{E}_P[Y_{\boldsymbol{x}_1} \mid \boldsymbol{x}_2, \boldsymbol{w}, \boldsymbol{c}]$. The last equality follows by the composition axiom of counterfactuals. Combining this expression with the decomposition of the policy effect we get,

$$\mathbb{E}_P[Y_{\boldsymbol{\pi}}] \geqslant \int \prod_{X \in \boldsymbol{X}} \pi(x \mid \boldsymbol{c}_X)P(\boldsymbol{c})\mathbb{E}_P[Y \mid \boldsymbol{x}_1, \boldsymbol{x}_2, \boldsymbol{w}, \boldsymbol{c}]P(\boldsymbol{x}_1 \mid \boldsymbol{x}_2, \boldsymbol{w}, \boldsymbol{c})P(\boldsymbol{w} \mid \boldsymbol{c}) \, d\boldsymbol{w} d\boldsymbol{c} d\boldsymbol{x}$$

$$= \int \bar{\boldsymbol{\pi}} y P(y \mid \boldsymbol{x}_1, \boldsymbol{x}_2, \boldsymbol{w}, \boldsymbol{c})P(\boldsymbol{x}_1 \mid \boldsymbol{x}_2, \boldsymbol{w}, \boldsymbol{c})P(\boldsymbol{w} \mid \boldsymbol{c})P(\boldsymbol{c}) \, dy d\boldsymbol{c} d\boldsymbol{x}$$

$$= \mathbb{E}_P[Y\bar{\boldsymbol{\pi}}/P(\boldsymbol{X}_2 \mid \boldsymbol{W}, \boldsymbol{C})].$$

For the upper bound, consider the following derivation,

$$\mathbb{E}_P[Y_{\boldsymbol{\pi}} \mid \boldsymbol{x}, \boldsymbol{c}]$$

$$= \int \mathbb{E}_P[Y_{\boldsymbol{x}_1} \mid \boldsymbol{x}_2, \boldsymbol{w}, \boldsymbol{c}] P(\boldsymbol{w} \mid \boldsymbol{c}) \, d\boldsymbol{w}$$

$$= \int \sum_{\boldsymbol{x}_1'} \{ \mathbb{E}_P[Y_{\boldsymbol{x}_1} \mid \boldsymbol{x}_1', \boldsymbol{x}_2, \boldsymbol{w}, \boldsymbol{c}] P(\boldsymbol{x}_1' \mid \boldsymbol{x}_2, \boldsymbol{w}, \boldsymbol{c}) \} P(\boldsymbol{w} \mid \boldsymbol{c}) \, d\boldsymbol{w}$$

$$= \int \Big\{ \mathbb{E}_P[Y_{\boldsymbol{x}_1} \mid \boldsymbol{x}_1, \boldsymbol{x}_2, \boldsymbol{w}, \boldsymbol{c}] P(\boldsymbol{x}_1 \mid \boldsymbol{x}_2, \boldsymbol{w}, \boldsymbol{c}) P(\boldsymbol{w} \mid \boldsymbol{c})$$
$$+ \sum_{\boldsymbol{x}_1' \neq \boldsymbol{x}_1} \mathbb{E}_P[Y_{\boldsymbol{x}_1} \mid \boldsymbol{x}_1', \boldsymbol{x}_2, \boldsymbol{w}, \boldsymbol{c}] P(\boldsymbol{x}_1' \mid \boldsymbol{x}_2, \boldsymbol{w}, \boldsymbol{c}) P(\boldsymbol{w} \mid \boldsymbol{c}) \Big\} \, d\boldsymbol{w}$$

$$\leqslant \int \Big\{ \mathbb{E}_P[Y \mid \boldsymbol{x}_1, \boldsymbol{x}_2, \boldsymbol{w}, \boldsymbol{c}] P(\boldsymbol{x}_1 \mid \boldsymbol{x}_2, \boldsymbol{w}, \boldsymbol{c}) P(\boldsymbol{w} \mid \boldsymbol{c}) + \sum_{\boldsymbol{x}_1' \neq \boldsymbol{x}_1} P(\boldsymbol{x}_1' \mid \boldsymbol{x}_2, \boldsymbol{w}, \boldsymbol{c}) P(\boldsymbol{w} \mid \boldsymbol{c}) \Big\} \, d\boldsymbol{w}$$

$$= \int \Big\{ \mathbb{E}_P[Y \mid \boldsymbol{x}_1, \boldsymbol{x}_2, \boldsymbol{w}, \boldsymbol{c}] P(\boldsymbol{x}_1 \mid \boldsymbol{x}_2, \boldsymbol{w}, \boldsymbol{c}) P(\boldsymbol{w} \mid \boldsymbol{c}) + (1 - P(\boldsymbol{x}_1 \mid \boldsymbol{x}_2, \boldsymbol{w}, \boldsymbol{c})) P(\boldsymbol{w} \mid \boldsymbol{c}) \Big\} \, d\boldsymbol{w}$$

$$= \int \mathbb{E}_P[Y \mid \boldsymbol{x}_1, \boldsymbol{x}_2, \boldsymbol{w}, \boldsymbol{c}] P(\boldsymbol{x}_1 \mid \boldsymbol{x}_2, \boldsymbol{w}, \boldsymbol{c}) P(\boldsymbol{w} \mid \boldsymbol{c}) \, d\boldsymbol{w} + 1 - \int P(\boldsymbol{x}_1 \mid \boldsymbol{x}_2, \boldsymbol{w}, \boldsymbol{c})) P(\boldsymbol{w} \mid \boldsymbol{c}) \, d\boldsymbol{w}.$$

The inequality follows from the boundedness of $Y$ and the rest of the arguments are analogous to the lower bound derivation. Combining this expression with the decomposition of the policy effect implies that,

$$\mathbb{E}_{P_{\boldsymbol{\pi}}}[Y] \leqslant \mathbb{E}_P[(Y - 1)\bar{\boldsymbol{\pi}}/P(\boldsymbol{X}_2 \mid \boldsymbol{W}, \boldsymbol{C})] + 1.$$

$\square$

**Prop. 4 restated**. *Let $\boldsymbol{Z}$ be an unconditional partial instrumental set with respect to a policy $\boldsymbol{\pi}$, i.e. $(Y \perp\!\!\!\perp_d \boldsymbol{Z})_{\mathcal{G}_{\boldsymbol{\pi}}}$. Then,*

$$\mathbb{E}_{P_{\boldsymbol{\pi}}}[Y] \geqslant \max_{\boldsymbol{z}} \mathbb{E}_P[Y\bar{\boldsymbol{\pi}} \mathbb{1}_{\boldsymbol{z}}(\boldsymbol{Z})/P(\boldsymbol{Z})]$$
$$\mathbb{E}_{P_{\boldsymbol{\pi}}}[Y] \leqslant \min_{\boldsymbol{z}} \mathbb{E}_P[(Y - 1)\bar{\boldsymbol{\pi}} \mathbb{1}_{\boldsymbol{z}}(\boldsymbol{Z})/P(\boldsymbol{Z})] + 1.$$

*Proof.* Consider the derivation of the lower bound. Let $\boldsymbol{\pi}$ denote a policy from an arbitrary set of covariates $\boldsymbol{C}$ to $\boldsymbol{X}$. Given that $(Y \perp\!\!\!\perp_d \boldsymbol{Z})_{\mathcal{G}_{\boldsymbol{\pi}}}$, by marginalizing over $\boldsymbol{X}, \boldsymbol{C}$ we find that

$$\mathbb{E}_{P_{\boldsymbol{\pi}}}[Y] = \mathbb{E}_P[Y_{\boldsymbol{\pi}}] = \mathbb{E}_P[Y_{\boldsymbol{\pi}} \mid \boldsymbol{z}] = \int \mathbb{E}_P[Y_{\boldsymbol{\pi}} \mid \boldsymbol{x}, \boldsymbol{c}, \boldsymbol{z}] \prod_{X \in \boldsymbol{X}} \pi_X(x \mid \boldsymbol{c}_X) P(\boldsymbol{c} \mid \boldsymbol{z}) \, d\boldsymbol{c} d\boldsymbol{x}.$$

$\boldsymbol{C}_X$ denotes the subset of $\boldsymbol{C}$ that is used to inform the intervention on $X$. Moreover,

$$\mathbb{E}_P[Y_{\boldsymbol{\pi}} \mid \boldsymbol{x}, \boldsymbol{c}, \boldsymbol{z}] \overset{(1)}{=} \mathbb{E}_P[Y_{\boldsymbol{x}} \mid \boldsymbol{c}, \boldsymbol{z}]$$
$$= \sum_{\boldsymbol{x}'} \mathbb{E}_P[Y_{\boldsymbol{x}} \mid \boldsymbol{x}', \boldsymbol{c}, \boldsymbol{z}] P(\boldsymbol{x}' \mid \boldsymbol{c}, \boldsymbol{z})$$
$$\geqslant \mathbb{E}_P[Y_{\boldsymbol{x}} \mid \boldsymbol{x}, \boldsymbol{c}, \boldsymbol{z}] P(\boldsymbol{x} \mid \boldsymbol{c}, \boldsymbol{z})$$
$$\overset{(2)}{=} \mathbb{E}_P[Y \mid \boldsymbol{x}, \boldsymbol{c}, \boldsymbol{z}] P(\boldsymbol{x} \mid \boldsymbol{c}, \boldsymbol{z}).$$

(1) holds by Rule 2 in Thm. 2 by swapping the policy $\boldsymbol{\pi}$ by the do intervention $do(\boldsymbol{X} = \boldsymbol{x})$: given that the policy acts on $\boldsymbol{X}$ taking as inputs $\boldsymbol{C}$ it is always true that $Y \perp\!\!\!\perp_d \boldsymbol{X} \mid \boldsymbol{C}$ in $\mathcal{G}_{\boldsymbol{\pi}\underline{\boldsymbol{X}}}$ and $\mathcal{G}_{\underline{\boldsymbol{X}\overline{\boldsymbol{X}}}}$ in any graph $\mathcal{G}$. The following equality follows by marginalizing over the domain of $\boldsymbol{X}$, which is assumed discrete. (2) follows by the composition axiom of counterfactuals. By combining these two

expressions we get

$$\mathbb{E}_P[Y_{\boldsymbol{\pi}}] \geqslant \int \prod_{X \in \boldsymbol{X}} \pi(x \mid \boldsymbol{c}_X) P(\boldsymbol{c} \mid \boldsymbol{z}) \mathbb{E}_P[Y \mid \boldsymbol{x}, \boldsymbol{c}, \boldsymbol{z}] P(\boldsymbol{x} \mid \boldsymbol{c}, \boldsymbol{z}) \, d\boldsymbol{c}d\boldsymbol{x}$$

$$= \int \bar{\pi} y P(y \mid \boldsymbol{x}, \boldsymbol{c}, \boldsymbol{z}) P(\boldsymbol{x} \mid \boldsymbol{c}, \boldsymbol{z}) P(\boldsymbol{c} \mid \boldsymbol{z}) \, dydcd\boldsymbol{x}$$

$$= \mathbb{E}_P[Y\bar{\pi} \mid \boldsymbol{z}]$$

$$= \mathbb{E}_P[Y\bar{\pi} \mathbb{1}_{\boldsymbol{z}}(\boldsymbol{Z})/P(\boldsymbol{Z})].$$

And since the l.h.s. does not depend on $\boldsymbol{Z}$ we can further tighten the bound by writing,

$$\mathbb{E}_P[Y_{\boldsymbol{\pi}}] \geqslant \max_{\boldsymbol{z}} \mathbb{E}_P[Y\bar{\pi} \mathbb{1}_{\boldsymbol{z}}(\boldsymbol{Z})/P(\boldsymbol{Z})].$$

Consider the derivation of the upper bound. It holds that,

$$\mathbb{E}_P[Y_{\boldsymbol{\pi}} \mid \boldsymbol{x}, \boldsymbol{c}, \boldsymbol{z}] = \mathbb{E}_P[Y_{\boldsymbol{x}} \mid \boldsymbol{c}, \boldsymbol{z}]$$

$$= \sum_{\boldsymbol{x}'} \mathbb{E}_P[Y_{\boldsymbol{x}} \mid \boldsymbol{x}', \boldsymbol{c}, \boldsymbol{z}] P(\boldsymbol{x}' \mid \boldsymbol{c}, \boldsymbol{z})$$

$$= \mathbb{E}_P[Y_{\boldsymbol{x}} \mid \boldsymbol{x}, \boldsymbol{c}, \boldsymbol{z}] P(\boldsymbol{x} \mid \boldsymbol{c}, \boldsymbol{z}) + \sum_{\boldsymbol{x}' \neq \boldsymbol{x}} \mathbb{E}_P[Y_{\boldsymbol{x}} \mid \boldsymbol{x}', \boldsymbol{c}, \boldsymbol{z}] P(\boldsymbol{x}' \mid \boldsymbol{c}, \boldsymbol{z})$$

$$\overset{(1)}{\leqslant} \mathbb{E}_P[Y \mid \boldsymbol{x}, \boldsymbol{c}, \boldsymbol{z}] P(\boldsymbol{x} \mid \boldsymbol{c}, \boldsymbol{z}) + \sum_{\boldsymbol{x}' \neq \boldsymbol{x}} P(\boldsymbol{x}' \mid \boldsymbol{c}, \boldsymbol{z})$$

$$= \mathbb{E}_P[Y \mid \boldsymbol{x}, \boldsymbol{c}, \boldsymbol{z}] P(\boldsymbol{x} \mid \boldsymbol{c}, \boldsymbol{z}) + 1 - P(\boldsymbol{x} \mid \boldsymbol{c}, \boldsymbol{z}).$$

(1) follows by the boundedness of $Y$, here assumed bounded by 1 for simplifying the derivation. As above by combining this inequality with the decomposition of the policy effect we get

$$\mathbb{E}_{P_{\boldsymbol{\pi}}}[Y] \leqslant \int \prod_{X \in \boldsymbol{X}} \pi(x \mid \boldsymbol{c}_X) P(\boldsymbol{c} \mid \boldsymbol{z}) \left\{ \mathbb{E}_P[Y \mid \boldsymbol{x}, \boldsymbol{c}, \boldsymbol{z}] P(\boldsymbol{x} \mid \boldsymbol{c}, \boldsymbol{z}) + 1 - P(\boldsymbol{x} \mid \boldsymbol{c}, \boldsymbol{z}) \right\} \, d\boldsymbol{c}d\boldsymbol{x}$$

$$= \mathbb{E}_P[Y\bar{\pi} \mid \boldsymbol{z}] + 1 - \mathbb{E}_P[\bar{\pi} \mid \boldsymbol{z}]$$

$$= \mathbb{E}_P[(Y - 1)\bar{\pi} \mid \boldsymbol{z}] + 1$$

$$= \mathbb{E}_P[(Y - 1)\bar{\pi} \mathbb{1}_{\boldsymbol{z}}(\boldsymbol{Z})/P(\boldsymbol{Z})] + 1.$$

And since the l.h.s. does not depend on $\boldsymbol{Z}$ we can further tighten the bound by writing,

$$\mathbb{E}_P[Y_{\boldsymbol{\pi}}] \geqslant \min_{\boldsymbol{z}} \mathbb{E}_P[(Y - 1)\bar{\pi} \mathbb{1}_{\boldsymbol{z}}(\boldsymbol{Z})/P(\boldsymbol{Z})] + 1.$$

$\square$

**Prop. 5 restated.** *Alg. 1 is sound.*

*Proof.* For the soundness of Alg. 1, we will consider each operation in turn and show that recovered adjustment sets and conditional instrumental sets lead to a valid bound.

*1. Omit redundant intervention variables.* For a policy $\boldsymbol{\pi} := \{\pi_X(\boldsymbol{c}_X)\}_{X \in \boldsymbol{X}}$ denote $\boldsymbol{\pi}_{\boldsymbol{R}} := \{\pi_X(\boldsymbol{c}_X)\}_{X \in \boldsymbol{R}}, \boldsymbol{R} \subseteq \boldsymbol{X}$. Let $\boldsymbol{R} = \boldsymbol{X} \cap An(Y)$ in $\mathcal{G}_{\boldsymbol{\pi}}$. By Rule 3 of the $\sigma$-calculus we have that,

$$\mathbb{E}_{P_{\boldsymbol{\pi}}}[Y] = \mathbb{E}_{P_{\boldsymbol{\pi}_{\boldsymbol{R}}}}[Y],$$

since $Y \perp\!\!\!\perp_d \boldsymbol{X} \backslash \boldsymbol{R}$ in $\mathcal{G}_{\boldsymbol{\pi}, \overline{\boldsymbol{X} \backslash \boldsymbol{R}}}$ and $\mathcal{G}_{\boldsymbol{\pi}_{\boldsymbol{R}}, \overline{\boldsymbol{X} \backslash \boldsymbol{R}}}$ as there are no directed paths from $\boldsymbol{X} \backslash \boldsymbol{R}$ to $Y$ by definition of $\boldsymbol{R}$. The first two lines therefore reduce the number of intervention variables. We proceed to bound the equivalent $\mathbb{E}_{P_{\boldsymbol{\pi}_{\boldsymbol{R}}}}[Y]$.

*2. Find partial adjustment sets $\boldsymbol{W}$.* Line 4-11 consider finding partial adjustment sets by traversing the set of treatment variables $R \in \boldsymbol{R}$. Recall that

$$\mathbb{E}_{P_{\boldsymbol{\pi}_{\boldsymbol{R}}}}[Y] = \mathbb{E}_P[Y_{\boldsymbol{\pi}_{\boldsymbol{R}}}] = \int \mathbb{E}_P[Y_{\boldsymbol{\pi}_{\boldsymbol{R}}} \mid \boldsymbol{r}, \boldsymbol{c}] \prod_{X \in \boldsymbol{R}} \pi_X(x \mid \boldsymbol{c}_X) P(\boldsymbol{c}) \, d\boldsymbol{c}d\boldsymbol{r}.$$

Denote $\bar{\pi}_R := \prod_{X \in R} \pi_X(x \mid c_X)$. In words lines 4-11 iteratively considers the existence of separators sets between $R \in \boldsymbol{R}$ and $Y$ to reduce the scope of the policy on $Y$, i.e. $Y_{\pi_R}$. By [43, Lemma 3.18.], if there exists a separator $\boldsymbol{W}$ between two sets of variables $\boldsymbol{X}$ and $\boldsymbol{Y}$ such that $\boldsymbol{S} \subseteq \boldsymbol{W} \subseteq \boldsymbol{K}$ in a graph $\mathcal{G}$, i.e. $\boldsymbol{X} \perp\!\!\!\perp_d \boldsymbol{Y} \mid \boldsymbol{W}$ in $\mathcal{G}$, then $\boldsymbol{W} = An(\boldsymbol{X} \cup \boldsymbol{Y} \cup \boldsymbol{S}) \cap \boldsymbol{K}$ is also a valid separator. The definition of $\boldsymbol{W}_R$ in line 7 aims to leverage this result to check for the existence of separators in our case.

Consider a $R \in \boldsymbol{R}$, let $\boldsymbol{S} = \boldsymbol{R} \backslash R$, $\boldsymbol{K} = \boldsymbol{V} \backslash (\boldsymbol{X} \cup \boldsymbol{C} \cup Y)$. In particular, if there exist a set $\{\boldsymbol{C}, \boldsymbol{R}\} \subset \boldsymbol{W} \subset \boldsymbol{K}$ such that $(Y \perp\!\!\!\perp_d R \mid \boldsymbol{W})$ in $\mathcal{G}_{\pi_{\boldsymbol{S}}, \underline{R}}$ then $\boldsymbol{W}_R = An(\boldsymbol{R} \cup Y \cup \boldsymbol{C}) \cap \boldsymbol{K} = An(Y \cup \boldsymbol{C}) \cap \boldsymbol{K}$ in $\mathcal{G}_{\pi_{\boldsymbol{S}}}$ satisfies $(Y \perp\!\!\!\perp_d R \mid \boldsymbol{W}_R)$ in $\mathcal{G}_{\pi_{\boldsymbol{S}}, \underline{R}}$, i.e. $\boldsymbol{W}_R$ is a separator if one exists. Note that $\boldsymbol{W}_R$ does not include any member of $\{\boldsymbol{C}, \boldsymbol{R}, Y\}$. Assume this separation holds and that the if statement in line 8 is triggered. Then,

$$\mathbb{E}_P[Y_{\pi_R} \mid \boldsymbol{r}, \boldsymbol{c}] = \mathbb{E}_P[Y_{\boldsymbol{r}} \mid \boldsymbol{c}]$$
$$= \mathbb{E}_P[Y_{\boldsymbol{s},r} \mid \boldsymbol{c}]$$
$$= \int \mathbb{E}_P[Y_{\boldsymbol{s},r} \mid \boldsymbol{c}, \boldsymbol{w}_R] P(\boldsymbol{w}_R \mid \boldsymbol{c}) \, d\boldsymbol{w}_R$$
$$= \int \mathbb{E}_P[Y_{\boldsymbol{s}} \mid r, \boldsymbol{c}, \boldsymbol{w}_R] P(\boldsymbol{w}_R \mid \boldsymbol{c}) \, d\boldsymbol{w}_R.$$

The last equality holds by assumption $(Y \perp\!\!\!\perp_d R \mid \boldsymbol{W}_R)$ in $\mathcal{G}_{\overline{\boldsymbol{S}}, \underline{R}}$.

Now considering the term $\mathbb{E}_P[Y_{\boldsymbol{s}} \mid r, \boldsymbol{c}, \boldsymbol{w}_R]$, for the second pass through the for loop, we can see that if the $d$-separation statement in the if condition is fulfilled we can further reduce the scope of the intervention on $\boldsymbol{S}$. In particular, for $R' \in \boldsymbol{R} \backslash R$, assuming that the if statement is triggered for $\boldsymbol{W}_{R'}$ we have that,

$$\mathbb{E}_P[Y_{\boldsymbol{s}} \mid r, \boldsymbol{c}, \boldsymbol{w}_R] = \int \mathbb{E}_P[Y_{\boldsymbol{s}} \mid r, \boldsymbol{c}, \boldsymbol{w}_R, \boldsymbol{w}_{R'}] P(\boldsymbol{w}_{R'} \mid \boldsymbol{w}_R, \boldsymbol{c}) \, d\boldsymbol{w}_{R'}$$
$$= \int \mathbb{E}_P[Y_{\boldsymbol{s} \backslash r'} \mid r', r, \boldsymbol{c}, \boldsymbol{w}_R, \boldsymbol{w}_{R'}] P(\boldsymbol{w}_{R'} \mid \boldsymbol{w}_R, \boldsymbol{c}) \, d\boldsymbol{w}_{R'}.$$

Upon reaching the end of the for loop we have for $\boldsymbol{W}$ defined in line 9, $\boldsymbol{T} = \{R : \boldsymbol{W}_R \in \boldsymbol{W}\}$, $\boldsymbol{U} = \boldsymbol{R} \backslash \boldsymbol{T}$ that,

$$\mathbb{E}_P[Y_{\boldsymbol{\pi}} \mid \boldsymbol{r}, \boldsymbol{c}] = \int \mathbb{E}_P[Y_{\boldsymbol{u}} \mid \boldsymbol{t}, \boldsymbol{c}, \boldsymbol{w}] P(\boldsymbol{w} \mid \boldsymbol{c}) \, d\boldsymbol{w}.$$

*3. Find partial instrumental sets $\boldsymbol{Z}$.* Starting line 12, we look for partial conditional instrumental sets. Consider first the case that a potential instruments $Z \in \boldsymbol{W}$ is evaluated in the for loop. On line 15, assume that the if statement is triggered, i.e. $(Y \perp\!\!\!\perp_d Z \mid \boldsymbol{W} \backslash Z, \boldsymbol{T})$ in $\mathcal{G}_{\overline{\boldsymbol{U}}}$ It then holds that,

$$\mathbb{E}_P[Y_{\boldsymbol{\pi}} \mid \boldsymbol{r}, \boldsymbol{c}] := \int \mathbb{E}_P[Y_{\boldsymbol{u}} \mid \boldsymbol{w}, \boldsymbol{t}, \boldsymbol{c}] P(\boldsymbol{w} \mid \boldsymbol{c}) \, d\boldsymbol{w}$$
$$= \int \mathbb{E}_P[Y_{\boldsymbol{u}} \mid \boldsymbol{w} \backslash z, z, \boldsymbol{t}, \boldsymbol{c}] \int_z P(\boldsymbol{w} \mid \boldsymbol{c}) \, d\boldsymbol{w}$$
$$= \int \mathbb{E}_P[Y_{\boldsymbol{u}} \mid \boldsymbol{w} \backslash z, z, \boldsymbol{t}, \boldsymbol{c}] P(\boldsymbol{w} \backslash z \mid \boldsymbol{c}) \, d\boldsymbol{w} \backslash z.$$

In turn, for the case that some $Z' \notin \boldsymbol{W}$ is evaluated in the for loop, if $(Y \perp\!\!\!\perp_d Z' \mid \boldsymbol{W}, \boldsymbol{T})$ in $\mathcal{G}_{\overline{\boldsymbol{U}}}$,

$$\mathbb{E}_P[Y_{\boldsymbol{\pi}} \mid \boldsymbol{r}, \boldsymbol{c}] := \int \mathbb{E}_P[Y_{\boldsymbol{u}} \mid \boldsymbol{w}, \boldsymbol{t}, \boldsymbol{c}] P(\boldsymbol{w} \mid \boldsymbol{c}) \, d\boldsymbol{w}$$
$$= \int \mathbb{E}_P[Y_{\boldsymbol{u}} \mid \boldsymbol{w}, \boldsymbol{t}, z', \boldsymbol{c}] P(\boldsymbol{w} \mid \boldsymbol{c}) \, d\boldsymbol{w}$$

For the $\boldsymbol{Z}$ obtained in line 15, we derive that,

$$\mathbb{E}_P[Y_{\boldsymbol{\pi}} \mid \boldsymbol{r}, \boldsymbol{c}] = \int \mathbb{E}_P[Y_{\boldsymbol{u}} \mid \boldsymbol{w} \backslash \boldsymbol{z}, \boldsymbol{z}, \boldsymbol{t}, \boldsymbol{c}] P(\boldsymbol{w} \backslash \boldsymbol{z} \mid \boldsymbol{c}) \, d\boldsymbol{w} \backslash \boldsymbol{z}.$$

*4. Return bounds.* We now proceed to use the expression above to lower and upper bound the effect of interest. In particular, for the lower bound we have that,

$$\mathbb{E}_P[Y_{\boldsymbol{\pi}} \mid \boldsymbol{x}, \boldsymbol{c}] = \int \mathbb{E}_P[Y_{\boldsymbol{u}} \mid \boldsymbol{w}\backslash\boldsymbol{z}, \boldsymbol{z}, \boldsymbol{t}, \boldsymbol{c}]P(\boldsymbol{w}\backslash\boldsymbol{z} \mid \boldsymbol{c})\,d\boldsymbol{w}\backslash\boldsymbol{z}$$

$$= \int \sum_{\boldsymbol{u}'} \mathbb{E}_P[Y_{\boldsymbol{u}} \mid \boldsymbol{u}', \boldsymbol{w}\backslash\boldsymbol{z}, \boldsymbol{z}, \boldsymbol{t}, \boldsymbol{c}]P(\boldsymbol{u}' \mid \boldsymbol{w}\backslash\boldsymbol{z}, \boldsymbol{z}, \boldsymbol{t}, \boldsymbol{c})P(\boldsymbol{w}\backslash\boldsymbol{z} \mid \boldsymbol{c})\,d\boldsymbol{w}\backslash\boldsymbol{z}$$

$$\geqslant \int \mathbb{E}_P[Y_{\boldsymbol{u}} \mid \boldsymbol{u}, \boldsymbol{w}\backslash\boldsymbol{z}, \boldsymbol{z}, \boldsymbol{t}, \boldsymbol{c}]P(\boldsymbol{u} \mid \boldsymbol{w}\backslash\boldsymbol{z}, \boldsymbol{z}, \boldsymbol{t}, \boldsymbol{c})P(\boldsymbol{w}\backslash\boldsymbol{z} \mid \boldsymbol{c})\,d\boldsymbol{w}\backslash\boldsymbol{z}$$

$$= \int \mathbb{E}_P[Y \mid \boldsymbol{u}, \boldsymbol{w}\backslash\boldsymbol{z}, \boldsymbol{z}, \boldsymbol{t}, \boldsymbol{c}]P(\boldsymbol{u} \mid \boldsymbol{w}\backslash\boldsymbol{z}, \boldsymbol{z}, \boldsymbol{t}, \boldsymbol{c})P(\boldsymbol{w}\backslash\boldsymbol{z} \mid \boldsymbol{c})\,d\boldsymbol{w}\backslash\boldsymbol{z}.$$

By combining the inequality above with the decomposition of the policy effect we get

$$\mathbb{E}_{P_{\boldsymbol{\pi}}}[Y]$$

$$\geqslant \int \prod_{X \in \boldsymbol{R}} \pi(x \mid \boldsymbol{c}_X)P(\boldsymbol{c})\mathbb{E}_P[Y \mid \boldsymbol{u}, \boldsymbol{w}\backslash\boldsymbol{z}, \boldsymbol{z}, \boldsymbol{t}, \boldsymbol{c}]P(\boldsymbol{u} \mid \boldsymbol{w}\backslash\boldsymbol{z}, \boldsymbol{z}, \boldsymbol{t}, \boldsymbol{c})P(\boldsymbol{w}\backslash\boldsymbol{z} \mid \boldsymbol{c})\,d\boldsymbol{c}d\boldsymbol{r}d\boldsymbol{w}\backslash\boldsymbol{z}$$

$$= \int \bar{\pi}_{\boldsymbol{R}} y P(y \mid \boldsymbol{r}, \boldsymbol{w}\backslash\boldsymbol{z}, \boldsymbol{z}, \boldsymbol{c})P(\boldsymbol{u} \mid \boldsymbol{w}\backslash\boldsymbol{z}, \boldsymbol{z}, \boldsymbol{t}, \boldsymbol{c})P(\boldsymbol{w}\backslash\boldsymbol{z} \mid \boldsymbol{c})P(\boldsymbol{c})\,d\boldsymbol{c}d\boldsymbol{r}d\boldsymbol{w}\backslash\boldsymbol{z}dy$$

$$= \int \bar{\pi}_{\boldsymbol{R}} y P(y \mid \boldsymbol{r}, \boldsymbol{w}\backslash\boldsymbol{z}, \boldsymbol{z}, \boldsymbol{c})P(\boldsymbol{u} \mid \boldsymbol{w}\backslash\boldsymbol{z}, \boldsymbol{z}, \boldsymbol{t}, \boldsymbol{c})\frac{P(\boldsymbol{t}, \boldsymbol{z} \mid \boldsymbol{w}\backslash\boldsymbol{z}, \boldsymbol{c})}{P(\boldsymbol{t}, \boldsymbol{z} \mid \boldsymbol{w}\backslash\boldsymbol{z}, \boldsymbol{c})}P(\boldsymbol{w}\backslash\boldsymbol{z} \mid \boldsymbol{c})P(\boldsymbol{c})\mathbb{1}_{\boldsymbol{z}}(\boldsymbol{z})\,d\boldsymbol{c}d\boldsymbol{r}d\boldsymbol{w}\backslash\boldsymbol{z}dydz$$

$$= \int \bar{\pi}_{\boldsymbol{R}} y P(y, \boldsymbol{r}, \boldsymbol{w}\backslash\boldsymbol{z}, \boldsymbol{z}, \boldsymbol{c})\frac{\mathbb{1}_{\boldsymbol{z}}(\boldsymbol{z})}{P(\boldsymbol{t}, \boldsymbol{z} \mid \boldsymbol{w}\backslash\boldsymbol{z}, \boldsymbol{c})}\,d\boldsymbol{c}d\boldsymbol{r}d\boldsymbol{w}\backslash\boldsymbol{z}dydz$$

$$= \mathbb{E}_P[Y\gamma]$$

where $\gamma := \bar{\pi}_{\boldsymbol{R}}\mathbb{1}_{\boldsymbol{z}}(\boldsymbol{Z})/P(\boldsymbol{T}, \boldsymbol{Z} \mid \boldsymbol{W}\backslash\boldsymbol{Z}, \boldsymbol{C})$. Note that we have used the definition $\boldsymbol{R} = \boldsymbol{T} \cup \boldsymbol{U}$.

In turn, for the upper bound we have that,

$$\mathbb{E}_P[Y_{\boldsymbol{\pi}} \mid \boldsymbol{x}, \boldsymbol{c}] = \int \mathbb{E}_P[Y_{\boldsymbol{u}} \mid \boldsymbol{w}\backslash\boldsymbol{z}, \boldsymbol{z}, \boldsymbol{t}, \boldsymbol{c}]P(\boldsymbol{w}\backslash\boldsymbol{z} \mid \boldsymbol{c})\,d\boldsymbol{w}\backslash\boldsymbol{z}$$

$$= \int \sum_{\boldsymbol{u}'} \mathbb{E}_P[Y_{\boldsymbol{u}} \mid \boldsymbol{u}', \boldsymbol{w}\backslash\boldsymbol{z}, \boldsymbol{z}, \boldsymbol{t}, \boldsymbol{c}]P(\boldsymbol{u}' \mid \boldsymbol{w}\backslash\boldsymbol{z}, \boldsymbol{z}, \boldsymbol{t}, \boldsymbol{c})P(\boldsymbol{w}\backslash\boldsymbol{z} \mid \boldsymbol{c})\,d\boldsymbol{w}\backslash\boldsymbol{z}$$

$$= \int \mathbb{E}_P[Y_{\boldsymbol{u}} \mid \boldsymbol{u}, \boldsymbol{w}\backslash\boldsymbol{z}, \boldsymbol{z}, \boldsymbol{t}, \boldsymbol{c}]P(\boldsymbol{u} \mid \boldsymbol{w}\backslash\boldsymbol{z}, \boldsymbol{z}, \boldsymbol{t}, \boldsymbol{c})P(\boldsymbol{w}\backslash\boldsymbol{z} \mid \boldsymbol{c})\,d\boldsymbol{w}\backslash\boldsymbol{z}$$

$$+ \int \sum_{\boldsymbol{u}' \neq \boldsymbol{u}} \mathbb{E}_P[Y_{\boldsymbol{u}} \mid \boldsymbol{u}', \boldsymbol{w}\backslash\boldsymbol{z}, \boldsymbol{z}, \boldsymbol{t}, \boldsymbol{c}]P(\boldsymbol{u}' \mid \boldsymbol{w}\backslash\boldsymbol{z}, \boldsymbol{z}, \boldsymbol{t}, \boldsymbol{c})P(\boldsymbol{w}\backslash\boldsymbol{z} \mid \boldsymbol{c})\,d\boldsymbol{w}\backslash\boldsymbol{z}$$

$$\leqslant \int \mathbb{E}_P[Y_{\boldsymbol{u}} \mid \boldsymbol{u}, \boldsymbol{w}\backslash\boldsymbol{z}, \boldsymbol{z}, \boldsymbol{t}, \boldsymbol{c}]P(\boldsymbol{u} \mid \boldsymbol{w}\backslash\boldsymbol{z}, \boldsymbol{z}, \boldsymbol{t}, \boldsymbol{c})P(\boldsymbol{w}\backslash\boldsymbol{z} \mid \boldsymbol{c})\,d\boldsymbol{w}\backslash\boldsymbol{z}$$

$$+ \int \sum_{\boldsymbol{u}' \neq \boldsymbol{u}} P(\boldsymbol{u}' \mid \boldsymbol{w}\backslash\boldsymbol{z}, \boldsymbol{z}, \boldsymbol{t}, \boldsymbol{c})P(\boldsymbol{w}\backslash\boldsymbol{z} \mid \boldsymbol{c})\,d\boldsymbol{w}\backslash\boldsymbol{z}$$

$$= \int \mathbb{E}_P[Y \mid \boldsymbol{u}, \boldsymbol{w}\backslash\boldsymbol{z}, \boldsymbol{z}, \boldsymbol{t}, \boldsymbol{c}]P(\boldsymbol{u} \mid \boldsymbol{w}\backslash\boldsymbol{z}, \boldsymbol{z}, \boldsymbol{t}, \boldsymbol{c})P(\boldsymbol{w}\backslash\boldsymbol{z} \mid \boldsymbol{c})\,d\boldsymbol{w}\backslash\boldsymbol{z}$$

$$+ \int (1 - P(\boldsymbol{u} \mid \boldsymbol{w}\backslash\boldsymbol{z}, \boldsymbol{z}, \boldsymbol{t}, \boldsymbol{c}))P(\boldsymbol{w}\backslash\boldsymbol{z} \mid \boldsymbol{c})\,d\boldsymbol{w}\backslash\boldsymbol{z}$$

$$= 1 + \int \mathbb{E}_P[Y - 1 \mid \boldsymbol{u}, \boldsymbol{w}\backslash\boldsymbol{z}, \boldsymbol{z}, \boldsymbol{t}, \boldsymbol{c}]P(\boldsymbol{u} \mid \boldsymbol{w}\backslash\boldsymbol{z}, \boldsymbol{z}, \boldsymbol{t}, \boldsymbol{c})P(\boldsymbol{w}\backslash\boldsymbol{z} \mid \boldsymbol{c})\,d\boldsymbol{w}\backslash\boldsymbol{z}$$

By combining the inequality above with the decomposition of the policy effect we get

$$\mathbb{E}_{P_{\boldsymbol{\pi}}}[Y]$$

$$\leqslant 1 + \int \prod_{X \in \boldsymbol{R}} \pi(x \mid \boldsymbol{c}_X) P(\boldsymbol{c}) \mathbb{E}_P[Y - 1 \mid \boldsymbol{u}, \boldsymbol{w} \backslash \boldsymbol{z}, \boldsymbol{z}, \boldsymbol{t}, \boldsymbol{c}] P(\boldsymbol{u} \mid \boldsymbol{w} \backslash \boldsymbol{z}, \boldsymbol{z}, \boldsymbol{t}, \boldsymbol{c}) P(\boldsymbol{w} \backslash \boldsymbol{z} \mid \boldsymbol{c}) \, d\boldsymbol{c} d\boldsymbol{r} d\boldsymbol{w} \backslash \boldsymbol{z}$$

$$= 1 + \int \bar{\pi}_{\boldsymbol{R}}(y - 1) P(y \mid \boldsymbol{r}, \boldsymbol{w} \backslash \boldsymbol{z}, \boldsymbol{z}, \boldsymbol{c}) P(\boldsymbol{u} \mid \boldsymbol{w} \backslash \boldsymbol{z}, \boldsymbol{z}, \boldsymbol{t}, \boldsymbol{c}) P(\boldsymbol{w} \backslash \boldsymbol{z} \mid \boldsymbol{c}) P(\boldsymbol{c}) \, d\boldsymbol{c} d\boldsymbol{r} d\boldsymbol{w} \backslash \boldsymbol{z} dy$$

$$= 1 + \int \bar{\pi}_{\boldsymbol{R}}(y - 1) P(y, \boldsymbol{r}, \boldsymbol{w} \backslash \boldsymbol{z}, \boldsymbol{z}, \boldsymbol{c}) \frac{\mathbb{1}_{\boldsymbol{z}}(\boldsymbol{z})}{P(\boldsymbol{t}, \boldsymbol{z} \mid \boldsymbol{w} \backslash \boldsymbol{z}, \boldsymbol{c})} \, d\boldsymbol{c} d\boldsymbol{r} d\boldsymbol{w} \backslash \boldsymbol{z} dy d\boldsymbol{z}$$

$$= 1 + \mathbb{E}_P[(Y - 1)\gamma]$$

where $\gamma := \bar{\pi}_{\boldsymbol{R}} \mathbb{1}_{\boldsymbol{z}}(\boldsymbol{Z}) / P(\boldsymbol{T}, \boldsymbol{Z} \mid \boldsymbol{W} \backslash \boldsymbol{Z}, \boldsymbol{C})$. Note that we have used the definition $\boldsymbol{R} = \boldsymbol{T} \cup \boldsymbol{U}$. $\quad\square$

**Prop. 6 restated.** *Let $k$ be the number of variables and $m$ be the number of edges in $\mathcal{G}$. The run time of Alg. 1 is $\mathcal{O}(k(k^2 + m))$.*

*Proof.* Let $k$ be the number of variables and $m$ be the number of edges in $\mathcal{G}$. Operations in Alg. 1, such as computing ancestors (e.g. lines 2,7), could be done in $\mathcal{O}(k^2)$ time, e.g. with a Breadth-First Search algorithm. Checking for $d$-separation is commonly done with the Bayes-Ball algorithm that can be implemented with a reachability search method, e.g. [43, Sec. 3.2.1], and could be done in $\mathcal{O}(k + m)$ time by [43, Prop. 3.17]. The sets $\boldsymbol{R}$ and $\boldsymbol{Z}$ have size at most $k$, so the for loops in lines 5 and 14 are executed at most $k$ times each. Combining these we get that Alg. 1 requires $\mathcal{O}(k(k^2 + m))$ time to return the bounds. $\quad\square$

**Prop. 7 restated.** *Let $\boldsymbol{\pi} := \{\pi_{\boldsymbol{X}_1}, \pi_{\boldsymbol{X}_2}\}$ be a policy on $\{\boldsymbol{X}_1, \boldsymbol{X}_2\}$ with a conditioning set $\boldsymbol{C}$. All partial adjustment sets may be enumerated in time $\mathcal{O}(k(k + m))$ where $k$ are the number of variables and $m$ be the number of edges in $\mathcal{G}$.*

*Proof.* This proposition is adapts Proposition 3.20 in [43] to find partial adjustment sets using the `ListSep` algorithm. `ListSep` performs backtracking to enumerate all separator sets $\boldsymbol{Z}$ between $\boldsymbol{X}$ and $\boldsymbol{Y}$ such that $\boldsymbol{I} \subseteq \boldsymbol{Z} \subseteq \boldsymbol{R}$, aborting branches that will not find a valid separator. In particular, it calls the `TestSep` and `FindSep` algorithms recursively.

The `TestSep` algorithm takes as input a graph, two sets of nodes to be separated, and a candidate separator set. By setting the graph to be $\mathcal{G}_{\pi_{\boldsymbol{X}_1} \underline{\boldsymbol{X}_2}}$, the two sets to be separated to be $\boldsymbol{Y}$ and $\boldsymbol{X}_2$, and the separator set $\boldsymbol{Z}$, `TestSep` will provably return whether $\boldsymbol{W} = \boldsymbol{Z} \backslash (\boldsymbol{C}, \boldsymbol{X}_1)$ is a partial adjustment set.

The `FindSep` algorithm uses the observation that if there exists a separator $\boldsymbol{Z}$ between $\boldsymbol{X}$ and $\boldsymbol{Y}$ such that $\boldsymbol{I} \subseteq \boldsymbol{Z} \subseteq \boldsymbol{R}$ then $\boldsymbol{Z} := An(\boldsymbol{X} \cup \boldsymbol{Y} \cup \boldsymbol{I}) \cap \boldsymbol{R}$ is a separator. For each possible partition $(\boldsymbol{X}_1, \boldsymbol{X}_2)$ of $\boldsymbol{X}$, we can therefore find a separator by testing (using `TestSep`) whether $An(\boldsymbol{X} \cup \boldsymbol{Y} \cup \boldsymbol{C}) \cap (\boldsymbol{V} \backslash (\boldsymbol{X} \cup \boldsymbol{Y}))$ in $\mathcal{G}_{\pi_{\boldsymbol{X}_1} \underline{\boldsymbol{X}_2}}$ is a separator. If it is then we have found a partial adjustment set.

To find all partial adjustment sets, we then proceed as follows. For each possible partition $(\boldsymbol{X}_1, \boldsymbol{X}_2)$ of $\boldsymbol{X}$, apply the `ListSep` algorithm with graph $\mathcal{G}_{\pi_{\boldsymbol{X}_1} \underline{\boldsymbol{X}_2}}$, sets to be separated $\boldsymbol{Y}$ and $\boldsymbol{X}_2$, and possible separator sets $\boldsymbol{Z}$ constrained as $\boldsymbol{C}, \boldsymbol{X}_1 \subseteq \boldsymbol{Z} \subseteq \boldsymbol{V} \backslash (\boldsymbol{X} \cup \boldsymbol{Y})$. Then, for every separator set $\boldsymbol{Z}$ found, output partial adjustment sets $\boldsymbol{W} = \boldsymbol{Z} \backslash (\boldsymbol{C}, \boldsymbol{X}_1)$.

This procedure finds all possible partial adjustment sets in time $\mathcal{O}(k(k + m))$ where $k$ are the number of variables and $m$ be the number of edges in $\mathcal{G}$ by Proposition 3.20 [43]. $\quad\square$

### B.2 Proofs of statements in Sec. 4

We make use of two auxiliary lemmas from the literature.

**Lemma 1** (Continuous Mapping Theorem, [26]). *Let $\{X_n\}_{n \in \mathbb{N}}$, $X$ be random elements defined on a metric space $S$. Consider a continuous function $g : S \to S'$ (where $S'$ is another metric space). Then,*

$$X_n \to_p X \Rightarrow g(X_n) \to_p g(X).$$

**Lemma 2** (Lemma 2, [24])**.** *Let $f_\eta := f(\boldsymbol{V}; \eta)$ denote a finite and continuous functional and $\eta$ denote its nuisances. For $n$ independent samples of $P$, $D := \{\boldsymbol{v}^{(i)}\}_{i=1,\ldots,n} \sim P$, let $\hat{T} = \mathbb{E}_D[f_{\hat\eta}]$ and $T := \mathbb{E}_P[f_\eta]$ for some $\eta$. Let $\mathbb{E}_{D-P}[f_\eta] := \mathbb{E}_D[f_\eta] - \mathbb{E}_P[f_\eta]$. Then, the following decomposition holds:*

$$\mathbb{E}_D[f_{\hat\eta}] - \mathbb{E}_P[f_\eta] = \underbrace{\mathbb{E}_{D-P}[f_\eta]}_{=A} + \underbrace{\mathbb{E}_{D-P}[f_{\hat\eta} - f_\eta]}_{=B} + \mathbb{E}_P[f_{\hat\eta} - f_\eta]. \tag{19}$$

*Suppose further that samples used for estimating $\eta$ are independent and separate; and the nuisances are consistent. Then, $R = A + B$ is a random variable that converges to zero at a rate $\mathcal{O}_P(1/\sqrt{n})$.*

**Prop. 8 restated.** *Suppose the nuisance estimates $(\hat{\boldsymbol{\mu}}, \hat{\boldsymbol{\gamma}})$ are $L_2$-consistent and bounded. Then, the error of the DML estimator $\hat{T}^{DML} \in \{\hat{T}^{DML,\ell}, \hat{T}^{DML,u}\}$ in Def. 5 is given as follows*

$$\hat{T}^{DML} - T^{DML} = \frac{1}{K}\sum_{k=1}^{K} R_k + O_P\left(\|\hat\gamma_{2,k} - \gamma_2\|\|\hat\mu_{2,k} - \mu_2\|\right) + O_P\left(\|\hat\gamma_{1,k} - \gamma_1\|\|\hat\mu_{1,k} - \hat{\tilde\mu}_{1,k}\|\right)$$

*where $R_k$ is a random variable that converges to zero at a rate $\mathcal{O}_P(1/\sqrt{n})$.*

*Proof.* In this proof we consider the estimation of the lower bund without loss of generality. Recall the definition of nuisance functions and estimators:

$$\gamma_2 := \bar\pi_U \gamma_1, \quad \gamma_1 := \bar\pi_T \mathbb{1}_{\boldsymbol{z}}(\boldsymbol{Z})/P(\boldsymbol{T}, \boldsymbol{Z} \mid \boldsymbol{W}\backslash\boldsymbol{Z}, \boldsymbol{C}),$$

with $T^{\text{PW},\ell} := \mathbb{E}_P[\gamma_2 Y] = \psi_{\boldsymbol{z}}^\ell$. And $\boldsymbol{\mu} = (\mu_0, \mu_1, \tilde\mu_2, \mu_2)$ defined by

$$\mu_2 := \mu_2(\boldsymbol{R}, \boldsymbol{C}, \boldsymbol{W}, \boldsymbol{Z}) = \mathbb{E}_P[Y \mid \boldsymbol{R}, \boldsymbol{C}, \boldsymbol{W}, \boldsymbol{Z}],$$
$$\tilde\mu_1 := \tilde\mu_2(\boldsymbol{R}, \boldsymbol{C}, \boldsymbol{W}, \boldsymbol{Z}) = \bar\pi_U(\boldsymbol{U} \mid \boldsymbol{C})\mu_2(\boldsymbol{R}, \boldsymbol{C}, \boldsymbol{W}, \boldsymbol{Z}),$$
$$\mu_1 := \mu_1(\boldsymbol{T}, \boldsymbol{C}, \boldsymbol{W}, \boldsymbol{Z}) = \mathbb{E}_P[\tilde\mu_2(\boldsymbol{R}, \boldsymbol{C}, \boldsymbol{W}, \boldsymbol{Z}) \mid \boldsymbol{T}, \boldsymbol{C}, \boldsymbol{W}, \boldsymbol{Z}],$$
$$\mu_0 := \mu_0(\boldsymbol{C}, \boldsymbol{W}, \boldsymbol{Z}) = \sum_t \mu_1(t, \boldsymbol{C}, \boldsymbol{W}, \boldsymbol{Z})\bar\pi_T(t \mid \boldsymbol{C}).$$

We first verify that the population level value of the regression estimator coincides with the lower bound $\psi_{\boldsymbol{z}}^\ell$. This can be seen with the following derivation,

$$T^{\text{REG},\ell} := \mathbb{E}_P[\mu_0(\boldsymbol{W}\backslash\boldsymbol{Z}, \boldsymbol{C}, \boldsymbol{z})]$$
$$= \sum_{\boldsymbol{w}\backslash\boldsymbol{z},\boldsymbol{c}} \mu_0(\boldsymbol{w}, \boldsymbol{c}, \boldsymbol{z})P(\boldsymbol{w}\backslash\boldsymbol{z}, \boldsymbol{c})$$
$$= \sum_{\boldsymbol{w}\backslash\boldsymbol{z},\boldsymbol{c}} \left(\sum_t \mu_1(t, \boldsymbol{w}, \boldsymbol{c}, \boldsymbol{z})\bar\pi_T\right)P(\boldsymbol{w}\backslash\boldsymbol{z}, \boldsymbol{c})$$
$$= \sum_{t,\boldsymbol{w}\backslash\boldsymbol{z},\boldsymbol{c}} \left(\sum_u \mu_2(t, u, \boldsymbol{c}, \boldsymbol{w}\backslash\boldsymbol{z}, \boldsymbol{z})\bar\pi_U P(u \mid t, \boldsymbol{z}, \boldsymbol{w}\backslash\boldsymbol{z}, \boldsymbol{c})\right)\frac{\bar\pi_T}{P(t, \boldsymbol{z} \mid \boldsymbol{w}\backslash\boldsymbol{z}, \boldsymbol{c})}P(t, \boldsymbol{z}, \boldsymbol{w}\backslash\boldsymbol{z}, \boldsymbol{c})$$
$$= \sum_{t,u,\boldsymbol{w}\backslash\boldsymbol{z},\boldsymbol{c}} \left(\sum_y yP(y \mid t, u, \boldsymbol{z}, \boldsymbol{w}\backslash\boldsymbol{z}, \boldsymbol{c})\right)\frac{\bar\pi_R}{P(t, \boldsymbol{z} \mid \boldsymbol{w}\backslash\boldsymbol{z}, \boldsymbol{c})}P(t, u, \boldsymbol{z}, \boldsymbol{w}, \boldsymbol{c})$$
$$= \sum_{y,t,u,\boldsymbol{z}',\boldsymbol{w}\backslash\boldsymbol{z}',\boldsymbol{c}} y\frac{\bar\pi_R\mathbb{1}_{\boldsymbol{z}}(\boldsymbol{z}')}{P(t, \boldsymbol{z}' \mid \boldsymbol{w}\backslash\boldsymbol{z}', \boldsymbol{c})}P(y, t, u, \boldsymbol{z}', \boldsymbol{w}\backslash\boldsymbol{z}', \boldsymbol{c})$$
$$= \mathbb{E}_P[Y\bar\pi_R\mathbb{1}_{\boldsymbol{z}}(\boldsymbol{Z})/P(\boldsymbol{T}, \boldsymbol{Z} \mid \boldsymbol{W}\backslash\boldsymbol{Z}, \boldsymbol{C})]$$
$$= \psi_{\boldsymbol{z}}^\ell$$

The DML estimator is similarly unbiased as,

$$T^{\text{DML},\ell} := \mathbb{E}_P[\gamma_2\{Y - \mu_2\}] + \mathbb{E}_P[\gamma_1\{\tilde\mu_1 - \mu_1\}] + \mathbb{E}_P[\mu_0]$$
$$= \mathbb{E}_P[\gamma_2\{\mathbb{E}_P[Y \mid \boldsymbol{R}, \boldsymbol{W}\backslash\boldsymbol{Z}, \boldsymbol{C}, \boldsymbol{Z}] - \mu_2\}] + \mathbb{E}_P[\gamma_1\{\tilde\mu_1 - \mu_1\}] + \mathbb{E}_P[\mu_0]$$
$$= \mathbb{E}_P[\gamma_2\{\mu_2 - \mu_2\}] + \mathbb{E}_P[\gamma_1\{\mathbb{E}_P[\tilde\mu_1 \mid \boldsymbol{T}, \boldsymbol{C}, \boldsymbol{W}\backslash\boldsymbol{Z}, \boldsymbol{Z}] - \mu_1\}] + \mathbb{E}_P[\mu_0]$$
$$= \mathbb{E}_P[\gamma_1\{\mu_1 - \mu_1\}] + \mathbb{E}_P[\mu_0]$$
$$= \mathbb{E}_P[\mu_0]$$
$$= \psi_{\boldsymbol{z}}^\ell.$$

Consider the estimated value of $T^{\mathrm{DML},\ell}$ following the procedure in Def. 5,

$$\hat{T}^{\mathrm{DML},\ell} := \frac{1}{K}\sum_{k=1}^{K}\hat{T}_k^{\mathrm{DML},\ell}, \quad \hat{T}_k^{\mathrm{DML},\ell} := \mathbb{E}_{\mathcal{D}^{(k)}}[\hat{\gamma}_2\{Y-\hat{\mu}_2\}] + \mathbb{E}_{\mathcal{D}^{(k)}}[\hat{\gamma}_1\{\hat{\tilde{\mu}}_1-\hat{\mu}_1\}] + \mathbb{E}_{\mathcal{D}^{(k)}}[\hat{\mu}_0],$$

We could then write,

$$\hat{T}_k^{\mathrm{DML},\ell} - T^{\mathrm{DML},\ell} = A + B + C$$

where,

$$A = \mathbb{E}_{\mathcal{D}^{(k)}-P}\Big[\gamma_2\{Y-\mu_2\} + \gamma_1\{\tilde{\mu}_1-\mu_1\} + \mu_0\Big]$$

$$B = \mathbb{E}_{\mathcal{D}^{(k)}-P}\Big[\Big(\gamma_2\{Y-\mu_2\}] + \gamma_1\{\tilde{\mu}_1-\mu_1\} + \mu_0\Big) - \Big(\hat{\gamma}_2\{Y-\hat{\mu}_2\} + \hat{\gamma}_1\{\hat{\tilde{\mu}}_1-\hat{\mu}_1\} + \hat{\mu}_0\Big)\Big]$$

$$C = \mathbb{E}_P\Big[\Big(\gamma\{Y-\mu_2\}] + \gamma\{\tilde{\mu}_1-\mu_1\} + \mu_0\Big) - \Big(\hat{\gamma}_2\{Y-\hat{\mu}_2\} + \hat{\gamma}_1\{\hat{\tilde{\mu}}_1-\hat{\mu}_1\} + \hat{\mu}_0\Big)\Big]$$

By the construction in Def. 5, the samples used for estimating $(\gamma_1,\gamma_2,\mu_0,\mu_1,\tilde{\mu}_1,\mu_2)$ and evaluating the outer expectation are independent and separate. Under the assumption that nuisance parameters $(\hat{\gamma}_1,\hat{\gamma}_2,\hat{\mu}_0,\hat{\mu}_1,\hat{\tilde{\mu}}_1,\hat{\mu}_2)$ are consistent, $R = A + B$ converges to zero at a rate $\mathcal{O}_P(1/\sqrt{|\mathcal{D}^{(k)}|})$ by Lem. 2.

Before manipulating $C$ and deriving its large sample behaviour, consider the following intermediate results:

$$\mathbb{E}_P[\mu_0] = \sum_{\boldsymbol{c},\boldsymbol{w}}\mu_0(\boldsymbol{c},\boldsymbol{w},\boldsymbol{z})P(\boldsymbol{c},\boldsymbol{w})$$

$$= \sum_{\boldsymbol{c},\boldsymbol{w}}\sum_{\boldsymbol{t}}\bar{\pi}_{\boldsymbol{T}}\mu_1(\boldsymbol{t},\boldsymbol{c},\boldsymbol{w},\boldsymbol{z})P(\boldsymbol{c},\boldsymbol{w})$$

$$= \sum_{\boldsymbol{c},\boldsymbol{w},\boldsymbol{z}',\boldsymbol{t}}\bar{\pi}_{\boldsymbol{T}}\mathbb{1}_{\boldsymbol{z}}(\boldsymbol{z}')\mu_1(\boldsymbol{t},\boldsymbol{c},\boldsymbol{w},\boldsymbol{z}')P(\boldsymbol{c},\boldsymbol{w})$$

$$= \sum_{\boldsymbol{c},\boldsymbol{w},\boldsymbol{z}',\boldsymbol{t}}\frac{\bar{\pi}_{\boldsymbol{T}}\mathbb{1}_{\boldsymbol{z}}(\boldsymbol{z}')}{P(\boldsymbol{t},\boldsymbol{z}'\mid\boldsymbol{c},\boldsymbol{w})}\mu_1(\boldsymbol{t},\boldsymbol{c},\boldsymbol{w},\boldsymbol{z}')P(\boldsymbol{t},\boldsymbol{c},\boldsymbol{w},\boldsymbol{z}')$$

$$= \mathbb{E}_P[\gamma_1\mu_1],$$

and similarly $\mathbb{E}_P[\hat{\mu}_0] = \mathbb{E}_P[\gamma_1\hat{\mu}_1]$. Note further that,

$$\mathbb{E}_P[\gamma_2\mu_2] = \sum_{\boldsymbol{c},\boldsymbol{w},\boldsymbol{t},\boldsymbol{u},\boldsymbol{z}'}\frac{\bar{\pi}_{\boldsymbol{T}}\bar{\pi}_{\boldsymbol{U}}\mathbb{1}_{\boldsymbol{z}}(\boldsymbol{z}')}{P(\boldsymbol{t},\boldsymbol{z}'\mid\boldsymbol{c},\boldsymbol{w})}\mu_2(\boldsymbol{c},\boldsymbol{w},\boldsymbol{t},\boldsymbol{u},\boldsymbol{z}')P(\boldsymbol{c},\boldsymbol{w},\boldsymbol{t},\boldsymbol{u},\boldsymbol{z}')$$

$$= \sum_{\boldsymbol{c},\boldsymbol{w},\boldsymbol{t},\boldsymbol{u},\boldsymbol{z}'}\gamma_1\bar{\pi}_{\boldsymbol{U}}\mu_2(\boldsymbol{c},\boldsymbol{w},\boldsymbol{t},\boldsymbol{u},\boldsymbol{z}')P(\boldsymbol{c},\boldsymbol{w},\boldsymbol{t},\boldsymbol{u},\boldsymbol{z}')$$

$$= \sum_{\boldsymbol{c},\boldsymbol{w},\boldsymbol{t},\boldsymbol{u},\boldsymbol{z}'}\gamma_1\tilde{\mu}_1(\boldsymbol{c},\boldsymbol{w},\boldsymbol{t},\boldsymbol{u},\boldsymbol{z}')P(\boldsymbol{c},\boldsymbol{w},\boldsymbol{t},\boldsymbol{z}')$$

$$= \mathbb{E}_P[\gamma_1\tilde{\mu}_1],$$

and similarly $\mathbb{E}_P[\gamma_2\hat{\mu}_2] = \mathbb{E}_P[\gamma_1\hat{\tilde{\mu}}_1]$.

Now consider $C = C_1 + C_2 + C_3$ where,

$$C_1 := \mathbb{E}_P[\hat{\gamma}_2\{Y-\hat{\mu}_2\} - \gamma_2\{Y-\mu_2\}] = \mathbb{E}_P[\hat{\gamma}_2\{\mu_2-\hat{\mu}_2\}]$$

$$C_2 := \mathbb{E}_P[\hat{\gamma}_1\{\hat{\tilde{\mu}}_1-\hat{\mu}_1\} - \gamma_1\{\tilde{\mu}_1-\mu_1\}]$$

$$C_3 := \mathbb{E}_P[\hat{\mu}_0-\mu_0] = \mathbb{E}_P[\gamma_1\hat{\mu}_1-\gamma_1\mu_1]$$

where the last equality follows from the result above. Therefore

$$
\begin{aligned}
C &= \mathbb{E}_P\Big[\hat{\gamma}_2\{\mu_2 - \hat{\mu}_2\} + \hat{\gamma}_1\{\hat{\tilde{\mu}}_1 - \hat{\mu}_1\} - \gamma_1\{\tilde{\mu}_1 - \mu_1\} + \gamma_1(\hat{\mu}_1 - \mu_1)\Big] \\
&= \mathbb{E}_P\Big[\{\hat{\gamma}_2 - \gamma_2\}\{\mu_2 - \hat{\mu}_2\} + \gamma_2\{\mu_2 - \hat{\mu}_2\} + \hat{\gamma}_1\{\hat{\tilde{\mu}}_1 - \hat{\mu}_1\} - \gamma_1\{\tilde{\mu}_1 - \mu_1\} + \gamma_1(\hat{\mu}_1 - \mu_1)\Big] \\
&= \mathbb{E}_P\Big[\{\hat{\gamma}_2 - \gamma_2\}\{\mu_2 - \hat{\mu}_2\}\Big] + \mathbb{E}_P\Big[\gamma_2\{\mu_2 - \hat{\mu}_2\} + \hat{\gamma}_1\{\hat{\tilde{\mu}}_1 - \hat{\mu}_1\} - \gamma_1\{\tilde{\mu}_1 - \hat{\mu}_1\}\Big] \\
&\overset{(1)}{=} \mathbb{E}_P\Big[\{\hat{\gamma}_2 - \gamma_2\}\{\mu_2 - \hat{\mu}_2\}\Big] + \mathbb{E}_P\Big[\gamma_1\{\tilde{\mu}_1 - \hat{\tilde{\mu}}_1\} + \hat{\gamma}_1\{\hat{\tilde{\mu}}_1 - \hat{\mu}_1\} - \gamma_1\{\tilde{\mu}_1 - \hat{\mu}_1\}\Big] \\
&= \mathbb{E}_P\Big[\{\hat{\gamma}_2 - \gamma_2\}\{\mu_2 - \hat{\mu}_2\}\Big] + \mathbb{E}_P\Big[-\gamma_1\{\hat{\tilde{\mu}}_1 - \hat{\mu}_1\} + \hat{\gamma}_1\{\hat{\tilde{\mu}}_1 - \hat{\mu}_1\}\Big] \\
&= O_P\Big(\|\hat{\gamma}_2 - \gamma_2\|\|\mu_2 - \hat{\mu}_2\|\Big) + O_P\Big(\|\hat{\gamma}_1 - \gamma_1\|\|\hat{\tilde{\mu}}_1 - \hat{\mu}_1\|\Big)
\end{aligned}
$$

(1) holds by the equalities $\mathbb{E}_P[\gamma_2\mu_2] = \mathbb{E}_P[\gamma_1\tilde{\mu}_1], \mathbb{E}_P[\gamma_2\hat{\mu}_2] = \mathbb{E}_P[\gamma_1\hat{\tilde{\mu}}_1]$.

Finally, this implies that

$$
\hat{T}^{\mathrm{DML},\ell} - T^{\mathrm{DML},\ell} = R + O_P\Big(\|\hat{\gamma}_2 - \gamma_2\|\|\mu_2 - \hat{\mu}_2\|\Big) + O_P\Big(\|\hat{\gamma}_1 - \gamma_1\|\|\hat{\tilde{\mu}}_1 - \hat{\mu}_1\|\Big) \qquad (20)
$$

where $R$ is a random variable that converges to zero at a rate $\mathcal{O}_P(1/\sqrt{n})$.

For the upper bound, consider the following definition of nuisance functions and estimators:

$$
\gamma_2 := \bar{\pi}_{\boldsymbol{U}}\gamma_1, \quad \gamma_1 := \bar{\pi}_{\boldsymbol{T}}\mathbb{1}_{\boldsymbol{z}}(\boldsymbol{Z})/P(\boldsymbol{T}, \boldsymbol{Z} \mid \boldsymbol{W}\backslash\boldsymbol{Z}, \boldsymbol{C}),
$$

with $T^{\mathrm{PW},\ell} := \mathbb{E}_P[\gamma_2(Y-1)] + 1 = \psi_{\boldsymbol{z}}^{\ell}$. And $\boldsymbol{\mu} = (\mu_0, \mu_1, \tilde{\mu}_2, \mu_2)$ defined by

$$
\begin{aligned}
\mu_2 &:= \mu_2(\boldsymbol{R}, \boldsymbol{C}, \boldsymbol{W}, \boldsymbol{Z}) = \mathbb{E}_P[Y - 1 \mid \boldsymbol{R}, \boldsymbol{C}, \boldsymbol{W}, \boldsymbol{Z}], \\
\tilde{\mu}_1 &:= \tilde{\mu}_2(\boldsymbol{R}, \boldsymbol{C}, \boldsymbol{W}, \boldsymbol{Z}) = \bar{\pi}_{\boldsymbol{U}}(\boldsymbol{U} \mid \boldsymbol{C})\mu_2(\boldsymbol{R}, \boldsymbol{C}, \boldsymbol{W}, \boldsymbol{Z}), \\
\mu_1 &:= \mu_1(\boldsymbol{T}, \boldsymbol{C}, \boldsymbol{W}, \boldsymbol{Z}) = \mathbb{E}_P[\tilde{\mu}_2(\boldsymbol{R}, \boldsymbol{C}, \boldsymbol{W}, \boldsymbol{Z}) \mid \boldsymbol{T}, \boldsymbol{C}, \boldsymbol{W}, \boldsymbol{Z}], \\
\mu_0 &:= \mu_0(\boldsymbol{C}, \boldsymbol{W}, \boldsymbol{Z}) = \sum_{\boldsymbol{t}} \mu_1(\boldsymbol{t}, \boldsymbol{C}, \boldsymbol{W}, \boldsymbol{Z})\bar{\pi}_{\boldsymbol{T}}(\boldsymbol{t} \mid \boldsymbol{C}).
\end{aligned}
$$

We first verify that the population level value of the regression estimator coincides with the lower bound $\psi_{\boldsymbol{z}}^{\ell}$. This can be seen with the following derivation,

$$
\begin{aligned}
T^{\mathrm{REG},u} &:= \mathbb{E}_P[\mu_0(\boldsymbol{W}\backslash\boldsymbol{Z}, \boldsymbol{C}, \boldsymbol{z})] + 1 \\
&= 1 + \sum_{\boldsymbol{w}\backslash\boldsymbol{z}, \boldsymbol{c}} \mu_0(\boldsymbol{w}, \boldsymbol{c}, \boldsymbol{z})P(\boldsymbol{w}\backslash\boldsymbol{z}, \boldsymbol{c}) \\
&= 1 + \sum_{\boldsymbol{w}\backslash\boldsymbol{z}, \boldsymbol{c}} \left(\sum_{\boldsymbol{t}} \mu_1(\boldsymbol{t}, \boldsymbol{w}, \boldsymbol{c}, \boldsymbol{z})\bar{\pi}_{\boldsymbol{T}}\right) P(\boldsymbol{w}\backslash\boldsymbol{z}, \boldsymbol{c}) \\
&= 1 + \sum_{\boldsymbol{t}, \boldsymbol{w}\backslash\boldsymbol{z}, \boldsymbol{c}} \left(\sum_{u} \mu_2(\boldsymbol{t}, \boldsymbol{u}, \boldsymbol{c}, \boldsymbol{w}\backslash\boldsymbol{z}, \boldsymbol{z})\bar{\pi}_{\boldsymbol{U}}P(\boldsymbol{u} \mid \boldsymbol{t}, \boldsymbol{z}, \boldsymbol{w}\backslash\boldsymbol{z}, \boldsymbol{c})\right) \frac{\bar{\pi}_{\boldsymbol{T}}}{P(\boldsymbol{t}, \boldsymbol{z} \mid \boldsymbol{w}\backslash\boldsymbol{z}, \boldsymbol{c})} P(\boldsymbol{t}, \boldsymbol{z}, \boldsymbol{w}\backslash\boldsymbol{z}, \boldsymbol{c}) \\
&= 1 + \sum_{\boldsymbol{t}, \boldsymbol{u}, \boldsymbol{w}\backslash\boldsymbol{z}, \boldsymbol{c}} \left(\sum_{y}(y - 1)P(y \mid \boldsymbol{t}, \boldsymbol{u}, \boldsymbol{z}, \boldsymbol{w}\backslash\boldsymbol{z}, \boldsymbol{c})\right) \frac{\bar{\pi}_{\boldsymbol{R}}}{P(\boldsymbol{t}, \boldsymbol{z} \mid \boldsymbol{w}\backslash\boldsymbol{z}, \boldsymbol{c})} P(\boldsymbol{t}, \boldsymbol{u}, \boldsymbol{z}, \boldsymbol{w}, \boldsymbol{c}) \\
&= 1 + \sum_{y, \boldsymbol{t}, \boldsymbol{u}, \boldsymbol{z}', \boldsymbol{w}\backslash\boldsymbol{z}', \boldsymbol{c}} (y - 1)\frac{\bar{\pi}_{\boldsymbol{R}}\mathbb{1}_{\boldsymbol{z}}(\boldsymbol{z}')}{P(\boldsymbol{t}, \boldsymbol{z}' \mid \boldsymbol{w}\backslash\boldsymbol{z}', \boldsymbol{c})} P(y, \boldsymbol{t}, \boldsymbol{u}, \boldsymbol{z}', \boldsymbol{w}\backslash\boldsymbol{z}', \boldsymbol{c}) \\
&= 1 + \mathbb{E}_P[(Y - 1)\bar{\pi}_{\boldsymbol{R}}\mathbb{1}_{\boldsymbol{z}}(\boldsymbol{Z})/P(\boldsymbol{T}, \boldsymbol{Z} \mid \boldsymbol{W}\backslash\boldsymbol{Z}, \boldsymbol{C})] \\
&= \psi_{\boldsymbol{z}}^u
\end{aligned}
$$

The DML estimator is similarly unbiased as,

$$
\begin{aligned}
T^{\mathrm{DML},u} &:= \mathbb{E}_P[\gamma_2\{(Y - 1) - \mu_2\}] + \mathbb{E}_P[\gamma_1\{\tilde{\mu}_1 - \mu_1\}] + \mathbb{E}_P[\mu_0] + 1 \\
&= \mathbb{E}_P[\gamma_2\{\mathbb{E}_P[Y - 1 \mid \boldsymbol{R}, \boldsymbol{W}\backslash\boldsymbol{Z}, \boldsymbol{C}, \boldsymbol{Z}] - \mu_2\}] + \mathbb{E}_P[\gamma_1\{\tilde{\mu}_1 - \mu_1\}] + \mathbb{E}_P[\mu_0] + 1 \\
&= \mathbb{E}_P[\gamma_2\{\mu_2 - \mu_2\}] + \mathbb{E}_P[\gamma_1\{\mathbb{E}_P[\tilde{\mu}_1 \mid \boldsymbol{T}, \boldsymbol{C}, \boldsymbol{W}\backslash\boldsymbol{Z}, \boldsymbol{Z}] - \mu_1\}] + \mathbb{E}_P[\mu_0] + 1 \\
&= \mathbb{E}_P[\gamma_1\{\mu_1 - \mu_1\}] + \mathbb{E}_P[\mu_0] + 1 \\
&= \mathbb{E}_P[\mu_0] + 1 \\
&= \psi_{\boldsymbol{z}}^u.
\end{aligned}
$$

The arguments of the proof are now analogous to that of the lower bound. We can conclude therefore that,

$$\hat{T}^{\text{DML},u} - T^{\text{DML},u} = R + O_P\Big(\|\hat{\gamma}_2 - \gamma_2\|\|\mu_2 - \hat{\mu}_2\|\Big) + O_P\Big(\|\hat{\gamma}_1 - \gamma_1\|\|\hat{\tilde{\mu}}_1 - \hat{\mu}_1\|\Big). \qquad (21)$$

$\square$

**Prop. 9 restated.** *Suppose either $\hat{\gamma}_1 = \gamma_1$ or $\hat{\mu}_2 = \mu_2$ and that either $\hat{\gamma}_1 = \gamma_1$ or $\hat{\tilde{\mu}}_1 = \hat{\mu}_1$. Then, $\hat{T}^{DML} \in \{\hat{T}^{DML,\ell}, \hat{T}^{DML,u}\}$ is an unbiased estimator of the corresponding bound defined in Alg. 1.*

*Proof.* Consider the estimated value of $T^{\text{DML},\ell}$ following the procedure in Def. 5,

$$\hat{T}^{\text{DML},\ell} := \frac{1}{K}\sum_{k=1}^{K}\hat{T}_k^{\text{DML},\ell}, \quad \hat{T}_k^{\text{DML},\ell} := \mathbb{E}_{\mathcal{D}^{(k)}}\big[\hat{\gamma}_2\{Y - \hat{\mu}_2\}\big] + \mathbb{E}_{\mathcal{D}^{(k)}}\big[\hat{\gamma}_1\{\hat{\tilde{\mu}}_1 - \hat{\mu}_1\}\big] + \mathbb{E}_{\mathcal{D}^{(k)}}\big[\hat{\mu}_0\big],$$

Under the assumption that,

$$\mathbb{E}_P[\hat{T}^{\text{DML},\ell}] = \mathbb{E}_P[\hat{\gamma}_2\{Y - \hat{\mu}_2\}] + \mathbb{E}_P[\hat{\gamma}_1\{\hat{\tilde{\mu}}_1 - \hat{\mu}_1\}] + \mathbb{E}_P[\hat{\mu}_0]$$

The bias of the estimator is given by,

$$\mathbb{E}_P[\hat{T}^{\text{DML},\ell}] - T^{\text{DML},\ell}$$

$$= \mathbb{E}_P\Big[\Big(\gamma\{Y - \mu_2\}] + \gamma\{\tilde{\mu}_1 - \mu_1\} + \mu_0\Big) - \Big(\hat{\gamma}_2\{Y - \hat{\mu}_2\} + \hat{\gamma}_1\{\hat{\tilde{\mu}}_1 - \hat{\mu}_1\} + \hat{\mu}_0\Big)\Big]$$

$$= \mathbb{E}_P\Big[\{\hat{\gamma}_2 - \gamma_2\}\{\mu_2 - \hat{\mu}_2\}\Big] + \mathbb{E}_P\Big[\{\hat{\gamma}_1 - \gamma_1\}\{\hat{\tilde{\mu}}_1 - \hat{\mu}_1\}\Big]$$

The last equality follows from the derivation of (C) in the proof of Prop. 8.

If either $\hat{\gamma}_1 = \gamma_1$ or $\hat{\mu}_2 = \mu_2$ and that either $\hat{\gamma}_1 = \gamma_1$ or $\hat{\tilde{\mu}}_1 = \tilde{\mu}_1$ then this expression equals 0 and we have that,

$$\mathbb{E}_P[\hat{T}^{\text{DML},\ell}] - T^{\text{DML},\ell} = 0.$$

An analogous result holds for the upper bound. $\square$

# C    Details on experiments

This section provides details of the data generating mechanisms used for synthetic experiments, and implementations.

As described in the experimental section, for estimating nuisances $(\gamma, \mu)$ we used Gradient Boosting models for classification and regression where appropriate. We implemented the models using Python using the commands `GradientBoostingClassifier()` and `GradientBoostingRegressor()` using default hyperparameters. We systematically bound probability values in the interval $[0.01, 0.99]$ to avoid propagating potentially large approximating errors.

## C.1    Simulations

We make use of the following 4 data generating systems, summarized by the graphs in Fig. 4. In this section, we use the notation $\mathbb{1}\{X\}$ that equals 1 if the statement $\{X\}$ is true, and equal to 0 otherwise.

**Example 1.**    The data generating mechanism for the first scenario is given by:

$$U_Z, U_{XY}, U_C \sim \mathcal{N}(0, 1),$$
$$Z := f_Z(U_Z),$$
$$C := f_C(U_C),$$
$$X := f_X(Z, C, U_{XY}),$$
$$Y := f_Y(X, C, U_{XY}),$$

where,

$$f_Z(U_Z) := -1 \cdot \mathbb{1}\{U_Z \leqslant -0.5\} + 0 \cdot \mathbb{1}\{-0.5 < U_Z < 0.5\} + 1 \cdot \mathbb{1}\{U_Z \geqslant 0.5\},$$
$$f_C(U_C) := U_C,$$
$$f_X(C, U_{XY}) := \mathbb{1}\{U_{XY} > 1 + \exp\{0.75C + 0.5Z + 0.5\}\},$$
$$f_Y(X, C, U_{XY}) := (2X - 1) - 2 \cdot (2U_{XY} - 1)$$

The outcome $Y$ is then re-scaled to lie in the $[0, 1]$ interval. The target for estimation can be derived with Alg. 1 and equals, for the lower and upper bound respectively,

$$\max_{z \in \{-1, 0, 1\}} \mathbb{E}_P[\bar{\pi} \mathbb{1}_z(Z) Y / P(Z \mid C)], \qquad \min_{z \in \{-1, 0, 1\}} \mathbb{E}_P[\bar{\pi} \mathbb{1}_z(Z)(Y - 1) / P(Z \mid C)] + 1.$$

**Example 2.**    The data generating mechanism for the second scenario is given by:

$$U_{X_1 Y}, U_{X_2 W}, U_{X_2}, U_{X_1}, U_C, U_W \sim \mathcal{N}(0, 1),$$
$$C := f_C(U_C),$$
$$W := f_W(U_{X_2 W}),$$
$$X_1 := f_{X_1}(C, U_{X_1 Y}, U_{X_1}),$$
$$X_2 := f_{X_2}(U_{X_2 W}, U_{X_2}),$$
$$Y := f_Y(X_1, X_2, C, W, U_{X_1 Y}),$$

where,

$$f_C(U_C) := U_C,$$
$$f_W(U_W) := U_{X_2 W},$$
$$f_{X_1}(C, U_{X_1}, U_{X_1 Y}) := \mathbb{1}\{U_{X_1 Y} \cdot (1 + \exp\{0.75C + 0.5\}) > U_{X_1}\},$$
$$f_{X_2}(U_{X_2 W}, U_{X_2}) := \mathbb{1}\{U_{X_2 W} + U_{X_2} > 0\},$$
$$f_Y(X_1, X_2, C, W, U_{X_1 Y}) := (2X_1 - 1) \cdot (C + 1) + 2\sin(2C \cdot (2X_1 - 1))$$
$$- 2(2U_{X_1 Y} - 1)(1 + 0.5C) + X_2 + W.$$

The outcome $Y$ is then re-scaled to lie in the $[0, 1]$ interval. The target for estimation can be derived with Alg. 1 and equals, for the lower and upper bound respectively,

$$\mathbb{E}_P[\bar{\pi} Y / P(X_2 \mid W, C)], \qquad \mathbb{E}_P[\bar{\pi}(Y - 1) / P(X_2 \mid W, C)] + 1.$$

**Example 3.** The data generating mechanism for the third scenario is given by:

$$U_{X_1Y}, U_{X_2W}, U_{X_2}, U_{X_1}, U_C \sim \mathcal{N}(0,1),$$
$$\boldsymbol{W} := \{W_1, \ldots, W_{100}\} \text{ s.t. } W_i \sim \mathcal{N}(0,1), i = 1, \ldots, 100,$$
$$C := f_C(U_C),$$
$$X_1 := f_{X_1}(C, U_{X_1Y}, U_{X_1}),$$
$$X_2 := f_{X_2}(\boldsymbol{W}, U_{X_2}),$$
$$Y := f_Y(X_1, X_2, C, \boldsymbol{W}, U_{X_1Y}),$$

where,

$$f_C(U_C) := U_C,$$
$$f_{X_1}(C, U_{X_1Y}, U_{X_1}) := \mathbb{1}\{U_{X_1Y} \cdot (1 + \exp\{0.75C + 0.5\}) > U_{X_1}\},$$
$$f_{X_2}(\boldsymbol{W}, U_{X_2}) := \mathbb{1}\{W_1 + W_2 + U_{X_2} > 0\},$$
$$f_Y(X_1, X_2, C, \boldsymbol{W}, U_{X_1Y}) := (2X_1 - 1) \cdot (C + 1) + 2\sin(2C \cdot (2X_1 - 1))$$
$$- 2(2U_{X_1Y} - 1)(1 + 0.5C) + X_2 + \sum_{i=1,\ldots,100} W_i/100.$$

The outcome $Y$ is then re-scaled to lie in the $[0,1]$ interval. The target for estimation can be derived with Alg. 1 and equals, for the lower and upper bound respectively,

$$\mathbb{E}_P[\bar{\boldsymbol{\pi}} Y / P(X_2 \mid \boldsymbol{W}, C)], \qquad \mathbb{E}_P[\bar{\boldsymbol{\pi}}(Y-1)/P(X_2 \mid \boldsymbol{W}, C)] + 1.$$

**Example 4.** The data generating mechanism for the fourth scenario is given by:

$$U_Z, U_{X_1Y}, U_{X_2}, U_{X_1}, U_C, U_Z \sim \mathcal{N}(0,1),$$
$$Z := f_Z(U_Z)$$
$$\boldsymbol{W} := \{W_1, \ldots, W_5\} \text{ s.t. } W_i \sim \mathcal{N}(0,1), i = 1, \ldots, 5,$$
$$C := f_C(U_C),$$
$$X_1 := f_{X_1}(C, Z, U_{X_1Y}, U_{X_1}, Z),$$
$$X_2 := f_{X_2}(\boldsymbol{W}, U_{X_2}),$$
$$Y := f_Y(X_1, X_2, C, W, U_{X_1Y}),$$

where,

$$f_C(U_C) := U_C,$$
$$f_Z(U_Z) := \mathbb{1}\{U_Z > 0\},$$
$$f_{X_1}(U_{X_1Y}, U_{X_1}, Z, U_C) := \mathbb{1}\{U_{X_1Y} \cdot (1 + \exp\{0.75U_C + 0.5Z + 0.5\}) > U_{X_1}\},$$
$$f_{X_2}(\boldsymbol{W}, U_{X_2}) := \mathbb{1}\{W_1 + W_2 + U_{X_2} > 0\},$$
$$f_Y(X_1, X_2, C, \boldsymbol{W}, U_{X_1Y}) := (2X_1 - 1) \cdot (C + 1) + 2\sin(2C \cdot (2X_1 - 1))$$
$$- 2(2U_{X_1Y} - 1)(1 + 0.5C) + X_2 + W_1$$

The outcome $Y$ is then re-scaled to lie in the $[0,1]$ interval. The target for estimation can be derived with Alg. 1 and equals, for the lower and upper bound respectively,

$$\max_{z \in \{0,1\}} \mathbb{E}_P[\bar{\boldsymbol{\pi}} \mathbb{1}_z(Z) Y / P(Z, X_2 \mid \boldsymbol{W}, C)], \qquad \min_{z \in \{0,1\}} \mathbb{E}_P[\bar{\boldsymbol{\pi}} \mathbb{1}_z(Z)(Y-1)/P(Z, X_2 \mid \boldsymbol{W}, C)] + 1.$$

## C.2 Health Campaign Evaluation

The data was curated from anonymous from Colombia, Peru and Mexico, using a web platform [30]. The exact questions considered in the survey can be found in [30]. The authors made the data available under a Creative Commons license[5] and is currently hosted by Kaggle as a c.s.v file, accessible through the following link: `kaggle.com/code/mpwolke/obesity-levels-life-style/`.

---

[5] `creativecommons.org/licenses/by/4.0/`

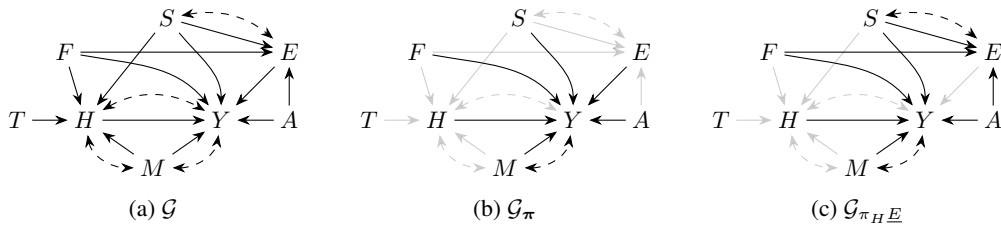

Figure 7: Obesity diagrams.

We considered a number of pre-processing steps. In particular, we created a BMI variable from weight and height data that were a posteriori removed from the dataset, imputed (with median or mode among observed instances) missing values, scaled continuously-valued covariates by subtracting the mean and dividing by the standard deviation, and scaled the outcome BMI to lie in the interval $[0, 1]$. Several covariates were ignored, including data on the consumption of water, consumption of vegetables, consumption of food between meals, consumption of alcohol, and number of main meals. These may be interpreted as unobserved variables, potentially confounding other relationships in the data as specified by the assumed causal diagram. The data from a total of 2111 individuals was recorded in this study.

The bounds on the effect of a policy $\pi^\alpha := \{\pi_H^\alpha, \pi_E^\alpha\}$ acting on $H$ and $E$, with new assignments defined as $\{P_{\text{new}}(H = \texttt{rarely}) = \alpha\}, P_{\text{new}}(E = \texttt{regularly}) = \alpha\}$ are approximated from the expression in Prop. 4. Following this proposition we find that Age $(A)$, Smoking $(S)$, calories consumption Monitoring $(M)$, and Family history with overweight $(F)$, that is $\boldsymbol{W} = \{A, S, M, F\}$, form a partial adjustment set for $\pi_E^\alpha$ given that $(Y \perp\!\!\!\perp_d E \mid \boldsymbol{W}, H)_{\mathcal{G}_{\pi_H E}}$. We can verify also that $\boldsymbol{Z} = \{T\}$ is a partial instrumental set for $\boldsymbol{\pi}$ since $(Y \perp\!\!\!\perp_d T)_{\mathcal{G}_{\boldsymbol{\pi}}}$. For illustration, the corresponding mutilated diagrams are shown in Fig. 7. Finally, we partition the data equally to obtain two sets of samples for training first and second stage classifiers and regressors, respectively, obtaining a first estimate of bounds, before then switching the role of the two data samples and averaging over the resulting estimates.

