# OpenReview forum: "Towards Estimating Bounds on the Effect of Policies under Unobserved Confounding"
_NeurIPS.cc/2024/Conference — NeurIPS 2024 poster_

### Official Review · Reviewer_nLc5 · 2024-07-06

**Soundness:** 2
**Presentation:** 2
**Contribution:** 2
**Rating:** 4
**Confidence:** 4

**Summary:**

The authors develop graphical characterizations and estimation tools to bound the effect of policies given a causal graph and observational data collected in non-identifiable settings.

**Strengths:**

Strengths and contributions: (1) derive analytical bounds for general probabilistic and conditional policies that are tighter than existing results, (2) develop an estimation framework to estimate bounds from finite samples, applicable in higher-dimensional spaces and continuously-valued data. They further show that the resulting estimators have favourable statistical properties such as fast convergence and robustness to model misspecification.

**Weaknesses:**

Please refer to Questions.

**Questions:**

Can the authors highlight the main challenge/novelty of the proposed bound for policy compared to bounds for treatment effect?

**Limitations:**

Please refer to Questions.

---

> ### Author Rebuttal · Authors · 2024-08-05
>
> Thank you for your review. Please consider the following descriptions and explanations to address your question.
>
> 1. ***”Can the authors highlight the main challenge/novelty of the proposed bound for policy compared to bounds for treatment effect?”***
>
> We would be happy to summarize what we consider to be the main contributions of our work. Below, we start by describing the context in which our methods are developed, followed by a description of the main novelty in our work.
>
> > Context (see also Sec. A.2 in the Appendix)
>
> Bounding the effect of interventions and policies, whenever they cannot be uniquely computed, is a research agenda that dates back to the work of Manski in the 1990's. Since, to produce better bounds, two lines of research have emerged making different assumptions on the underlying structural model. One the one hand making assumptions on the strength of unobserved confounding, and testing the sensitivity of conclusions as the strength of unobserved confounding increases. And, on the other hand, making assumptions on the structural dependencies between variables, encoded in a causal diagram.
>
> Our approach considers the second setting. In other words, we attempt to provide better bounds and estimation methods for the effect of policies given data and an (arbitrary) causal diagram of the system. Very few practical solutions exist for that problem. Most existing work has focused on solving constrained optimization problems in parameterized models as a way to bound the effect of interventions, e.g. [42, 12, 25, 13],
> \\[\min / \max_{\mathcal M_\theta} \mathbb E_{P_{\mathcal M_\theta}}[Y | do(x)], \quad \text{such that}\quad P_{\mathcal M_\theta}(\boldsymbol v)=P(\boldsymbol v).\\]
> In words, the problem is to search over the space of models $\mathcal M_\theta$ parameterized by $\theta$ for the minimum and maximum values of $\mathbb E_{P}[Y | do(x)]$ such that $\mathcal M_\theta$ entails the observed distribution, i.e. $P_{\mathcal M_\theta}(\boldsymbol v)=P(\boldsymbol v)$. The challenge with this approach is that the number of parameters $\theta$ grows exponentially with the number of variables and their cardinality, and is difficult to scale to high-dimensional systems or continuous variables without assumptions on the function class and distributional families.
>
> > Novelty
>
> In light of this summary, our first contribution is to derive analytical bounds (that is, mathematical expressions in terms of $P(\boldsymbol V)$) on the effect of policies that exploit the structure of the causal diagram. The proof strategy is based on Manski’s bounds (Prop. 1) but extends them to leverage the independencies between variables given the causal diagram, producing new graphical criteria and tighter bounds (Props. 2, 3, 4, and Alg. 1). In particular, the bounds hold in systems with continuously-valued outcomes and covariates – which is a distinction with respect to previous work. Our second contribution is to develop an estimation strategy that would allow for the practical computation of bounds in high-dimensional systems, and with favorable statistical guarantees such as robustness to misspecification and fast convergence.

---

> > ### Author Response · Authors · 2024-08-13
> >
> > With the discussion period coming to its end, we were wondering whether you had a chance to check our rebuttal. Please don't hesitate to get in touch if there is any concern we could still help to clarify.
> >
> > Thank you.

---

### Official Review · Reviewer_uSXJ · 2024-07-11

**Soundness:** 3
**Presentation:** 3
**Contribution:** 3
**Rating:** 7
**Confidence:** 3

**Summary:**

This paper proposes a partial identification method for the value of policies that intervene on a potentially multivariate set of variables in a known causal graph. It searches for adjustment sets or IVs that apply to one set of variables intervened on conditioned on others.

**Strengths:**

Expanding the reach of partial identification methods to account for new kinds of action/covariate spaces is a valuable contribution. The paper is well-executed, with clean algorithmic strategies and statistical guarantees for the double ML estimator. I also appreciate the application to evaluating health campaigns -- although the bounds obtained in this case are not incredibly tight, they show that the method could provide suggestive evidence in a setting of practical interest.

**Weaknesses:**

(1) It would be valuable to have more intuition about what the method is doing. My heuristic understanding right now is that it is combining bounds for policies that intervene on two sets of variables, one of which can be made to satisfy a no-unobserved-confounders type assumption given the graph, and one of which cannot (and hence we get fairly trivial bounds for that portion of the treatment variables). What would the authors say is the right way to understand the strategy? Similarly, it would be useful to contrast explicitly with, e.g., previous strategies for partial identification with discrete variables.

(2) The majority of the experiments are spent on evaluating accuracy of the DML method at estimating the bounds. This is valuable, but for partial identification methods the more important question is likely to be whether the bounds are informative. It would be more useful to have experiments that assess what kinds of graph structures lead to more or less informative bounds, to guide potential users about when the method is likely to be useful, even if it meant pushing some of the current experiments content to an appendix

**Questions:**

See (1) above.

**Limitations:**

Yes

---

> ### Author Rebuttal · Authors · 2024-08-05
>
> Thank you for your review and positive feedback. In the following we hope to address the mentioned weaknesses. Please let us know if we can expand on any of it.
>
> 1. ***”It would be valuable to have more intuition about what the method is doing. My heuristic understanding right now is that it is combining bounds for policies that intervene on two sets of variables, one of which can be made to satisfy a no-unobserved-confounders type assumption given the graph, and one of which cannot (and hence we get fairly trivial bounds for that portion of the treatment variables). What would the authors say is the right way to understand the strategy? Similarly, it would be useful to contrast explicitly with, e.g., previous strategies for partial identification with discrete variables.”***
>
> Thank you for this question. Intuitively, we would describe the method as seeking to recursively simplify the query. First by omitting the intervened variables that have no effect on the outcome; second by finding the set of intervened variables for which a partial adjustment set could be used to tighten the bound, and third by finding the set of variables that act as valid partial instrumental sets to evaluate bounds on the most favorable conditional distributions rather than on the joint distribution.
>
> The previous strategies for partial identification are described in Sec. A.2. We will refine that section to make a more explicit comparison. In our view, the biggest contrast with existing partial identification methods is that we provide analytical bounds (written in terms of $P(\boldsymbol v)$) rather than as solutions to a constrained optimization problem. The biggest advantage of our approach is that analytical bounds allow for efficient estimation in high-dimensional systems of continuously-valued variables with the proposed estimation framework (Sec. 4).
>
> 2. ***”The majority of the experiments are spent on evaluating accuracy of the DML method at estimating the bounds. This is valuable, but for partial identification methods the more important question is likely to be whether the bounds are informative. It would be more useful to have experiments that assess what kinds of graph structures lead to more or less informative bounds, to guide potential users about when the method is likely to be useful, even if it meant pushing some of the current experiments content to an appendix.”***
>
> This is an interesting point. In practice, although the graph structure can play an important role in tightening bounds, the actual width of the bounds in a particular problem are also driven by the distribution of data. Here is an example to make this more concrete.
>
> For binary variables $X$ and $Y$, consider bounding $P(Y=1 | do(X=1))$ from $P(X,Y)$ and the bow graph $\\{X \rightarrow Y, X\leftrightarrow Y\\}$. The tightest bounds in this case are given by:
>
> \\[P(Y=1,X=1) \leq P(Y=1 | do(X=1)) \leq P(Y=1,X=1) + 1 - P(X=1)\\]
>
> The width of the bounds is given by $1 - P(X=1)$. This term may evaluate to anything between $(0,1)$ depending on the value of $P(X=1)$.
>
> For more complex causal structures, the bounds proposed in Prop. 3 and Prop. 4 (and Alg. 1) reduce the width of the interval above. Intuitively, from $1 - P(X=1)$ to $1 - P(X=1)\alpha$ for some $\alpha\geq 1$ that is a function of the joint distribution. But again, the actual width of the interval is ultimately determined by the values of $\alpha$ and $P(X=1)$ that may be large or small depending on the value probabilities involved. A similar reasoning may be given for the effect of policies instead of the effect of hard interventions.

---

> > ### Author Response · Authors · 2024-08-13
> >
> > We hope to have answered all concerns to your satisfaction in the rebuttal above. Please don't hesitate to get in touch if there is any concern we could still help to clarify.
> >
> > Thank you again for the positive feedback.

---

### Official Review · Reviewer_BoZN · 2024-07-12

**Soundness:** 3
**Presentation:** 4
**Contribution:** 3
**Rating:** 7
**Confidence:** 4

**Summary:**

This paper proposes a new method to partially identify policy effects using a known causal graph and observational data. In theory, the paper derives general bounds for probabilistic and conditional policies, generalizing the existing results on discrete atomic policies. To estimate the bounds, the authors extend the well-known double machine learning framework to their setup. Finally, simulations are provided to demonstrate the convergence rate of the estimators. Then, the estimators are applied to estimate bounds for a range of policies in an obesity study.

**Strengths:**

The paper is overall well-written and organized. The contributions of the paper are clear in contrast with previous works. The authors not only derive new theoretical bounds but also provide an efficient double machine learning method to estimate the bounds. Finally, experiments are provided to demonstrate the robustness against the issue of model misspecification.

**Weaknesses:**

1. I think the paper can discuss a bit more about the cases in which the policy effect is identifiable, given a known graph and observational data, before moving to partial identification. Generally, partial identification is valuable when the assumption required for point identification may not hold in practice. The authors can potentially discuss the assumption and highlight the value of their method.
2. I didn't see any assumption required to derive the natural bound and its extension proposed by the authors. If the bound is assumption-free, would it be too conservative to use in practice? The authors should clarify this.
3. The estimation strategies in the paper are quite standard. The authors should explain why this part is novel compared to the existing literature.

**Questions:**

N.A.

**Limitations:**

N.A.

---

> ### Author Rebuttal · Authors · 2024-08-05
>
> Thank you for your thoughtful review and positive assessment of our work, we appreciate the feedback. Please find our response to specific concerns and questions in the following.
>
> 1. ***”I think the paper can discuss a bit more about the cases in which the policy effect is identifiable, given a known graph and observational data, before moving to partial identification. Generally, partial identification is valuable when the assumption required for point identification may not hold in practice. The authors can potentially discuss the assumption and highlight the value of their method.”***
>
> This is a good suggestion, thank you. Identification, and the type of graphs that enable identification of the effect of policies, will be mentioned more explicitly in the introductory sections.
>
> 2. ***”I didn't see any assumption required to derive the natural bound and its extension proposed by the authors. If the bound is assumption-free, would it be too conservative to use in practice? The authors should clarify this.”***
>
> The only requirement for the natural bounds (Props. 1 and 2) is for the treatment variables $\boldsymbol X$ to be discretely-valued and for $Y$ to be bounded.
>
> As the reviewer suggests, if more information is available, such as a causal diagram of the environment, the bounds can typically be improved by exploiting this structure, and the natural bounds will be too conservative. In our paper, we currently convey this idea in Example 2 that shows how the knowledge of the causal diagram leads to a bound tighter than the natural bounds. We will make sure that this is more clearly stated.
>
> 3. ***”The estimation strategies in the paper are quite standard. The authors should explain why this part is novel compared to the existing literature.”***
>
> The general principles behind the definition of double machine learning estimators have appeared in previous work. To a large extend we do build on previous intuition. In our view, however, adapting these techniques for the new expressions of the bounds in Alg. 1 is non-trivial. The nuisance functions, the definition of nested regressions, and target estimators, are all specifically tailored to our setting, and are novel contributions to the literature. Similarly for the corresponding robustness guarantee and rate of convergence result. We do believe that developing DML estimators for the bounds we derive and, as a consequence, providing a complete solution for estimating bounds on the effect of policies is novel and impactful.

---

> > ### Author Response · Authors · 2024-08-13
> >
> > The discussion period is reaching its end. We hope you have had the chance to check our rebuttal and wonder whether it has answered your questions. If not, we would be happy to expand on any remaining concerns.
> >
> > We appreciate your time and attention. Thank you!

---

### Official Review · Reviewer_LyZM · 2024-07-16

**Soundness:** 2
**Presentation:** 2
**Contribution:** 2
**Rating:** 5
**Confidence:** 2

**Summary:**

The paper proposes a new framework that uses structural causal graphs for estimating the effect of policies under unobserved confounding. Their approach first derives tighter analytical bounds on the estimates of the effects of the policies under the structural causal graph setting and the paper uses the results to develop estimators of these bounds using the double machine learning toolkit. The provide numerical experiments to show their approach produces good estimates of the effects of the policies under different settings such as misspecification.

**Strengths:**

The paper tackles an interesting problem of estimating the effect of policies under unobserved confounding and develops an effective structural causal models approach. The paper provides good discussion on the analytical bounds and is able to study multiple settings in the numerics.

**Weaknesses:**

The paper is somewhat hard to follow as the authors of the paper assume readers are familiar with existing notation for structural causal graphs. For example, they assume readers know the notation "an, de" for such graphs. Much of the notation is defined ahead of time, but without context for how and where they are used throughout the paper. Additionally, some new notation appear without explanation and are left to be interpreted by the reader. For example, in the partial identification section, $\bf{X}_1, \bf{X}_2$ are introduced, but the paper provides minimal intuition for what each set of variables means or should mean. As a result, the paper in general is hard to follow as it is overloaded with notation but limited intuition. Potentially, more concrete examples could be provided. The partial identification section in particular could benefit from figures of graphs that satisfy definition 3.

In the numerics, it may be helpful to develop some sort of baseline to demonstrate how the new estimation framework out performs existing approaches.

The paper also seems to have various typos and broken references. There are missing citations at line 67 and 311 and a missing proposition reference on line 316.

**Questions:**

1. Can more detailed be given on differences between $T^{TW}$ and $T^{REG}$ that can explain the differences in the numerics? Is $T^{DML}$ derived from one of the two estimators combined with the DML toolkit and if so which one is used?
2. Since the approaches provide an upper and lower bound on the effect of the policies, how do you select an estimate if the difference between the bounds is large?

**Limitations:**

Limitations have been addressed by authors.

---

> ### Author Rebuttal · Authors · 2024-08-05
>
> Thank you for your thoughtful review. The typos and missing / broken references have been corrected, thank you. In the following, we aim to address your concerns point by point.
>
> 1. ***“The paper is somewhat hard to follow as the authors of the paper assume readers are familiar with existing notation for structural causal graphs.***
>
> We appreciate the feedback on the readability of the paper. We will aim to improve following your suggestions but we do feel that (some of) our choices are well justified. Please consider the points below to help address specific concerns mentioned in the review.
>
> > "Much of the notation is defined ahead of time, but without context for how and where they are used throughout the paper."
>
> It is not uncommon to describe the notation used throughout the paper in a short, self-contained section at the beginning of a paper. In our case, we found that defining different mathematical symbols in the main body, where required, creates a disconnect with the overall story line that makes the paper less readable.
>
> > "Additionally, some new notation appear without explanation and are left to be interpreted by the reader. For example, in the partial identification section, $\boldsymbol X_1, \boldsymbol X_2$ are introduced, but the paper provides minimal intuition for what each set of variables means or should mean."
>
> Def. 3 mentions explicitly that $\boldsymbol X_1, \boldsymbol X_2$ are two sets of variables that are intervened on by the policy $\boldsymbol\pi$. We have also made an effort to introduce the intuition behind the partial adjustment strategy in Def. 3 gradually, first highlighting the idea in Example 2 and then again describing in words the proposition that uses Def. 3 in lines 166-167. Note also that Examples 3 and 4 both illustrate these ideas once more.
>
> > "Potentially, more concrete examples could be provided. The partial identification section in particular could benefit from figures of graphs that satisfy definition 3."
>
> We made a conscious effort to provide a concrete example after every proposition to illustrate the ideas involved. For example, Def. 3 is used and explicitly mentioned in Examples 3 and 4 using the graphs in Figs. 2b and 2c.
>
> 2. ***”In the numerics, it may be helpful to develop some sort of baseline to demonstrate how the new estimation framework out performs existing approaches.”***
>
> This suggestion is a very good one, thank you.
>
> We are considering an additional experiment to give a sense of the benefit of the progressively tighter bounds given in Sec. 3. In particular, we can compute each one of these bounds separately for a concrete example and inspect their values.
>
> For this experiment, we sample from the data generating mechanism associated with the causal graph given in the fourth row of Fig. 4, and evaluate the policy in Eq. (16). We compute bounds obtained by the natural bounds (most conservative, Prop. 2),  using the partial adjustment set $W$ only (Prop. 3), using the partial instrumental set $Z$ only (Prop. 4), and finally using Alg. 1 that combines all propositions and is the proposed approach. We display figures with the resulting bounds and causal graph in the attached pdf, demonstrating the gain that can be achieved in this particular example with our proposed bounding strategy.
>
> Finally, we would like to note that, to our knowledge, no existing method has been developed to estimate bounds on the effect of policies given an arbitrary causal diagram of the environment.
>
> 3. ***“Can more detailed be given on differences between PW and REG that can explain the differences in the numerics?"***
>
> Of course, thank you for the question. We could start by noting that $T^{PW}$ involves the estimation of a ratio of probabilities $\boldsymbol\\gamma$, while $T^{REG}$ is based on a sequence of regression tasks. $T^{PW}$ can therefore be unstable with low sample sizes if the ratio denominator is estimated to be close to zero. This might explain the relatively larger errors of $T^{PW}$ in Fig. 4 for low sample sizes. $T^{REG}$, in contrast, is better behaved in small sample sizes, outperforming $T^{PW}$.
> For equally good estimators of nuisance functions ($\boldsymbol\\gamma$ for $T^{PW}$ and $\boldsymbol\mu$ for $T^{REG}$), both estimators have rates of convergence of the same order. With larger samples, we expect them to converge to the true values at a similar rate.
>
> 4. ***"Is DML derived from one of the two estimators combined with the DML toolkit and if so which one is used?”***
>
> $T^{DML}$ is constructed as a combination of elements of $T^{PW}$ and $T^{REG}$. In particular, note in the definition of $T^{DML}$ (Def. 5) the use of nuisances $\boldsymbol\gamma$ of $T^{PW}$ and $\boldsymbol\mu$ of $T^{REG}$. As a result, $T^{DML}$ has quite a different performance profile than $T^{PW}$ or $T^{REG}$, being more robust to noise in nuisance estimation and converging at a faster rate to the underlying population-level bounds as a function of sample size.
>
> 5. ***“Since the approaches provide an upper and lower bound on the effect of the policies, how do you select an estimate if the difference between the bounds is large?”***
>
> This is an important point, thank you for raising it.
>
> By the definition of bounds, the true value of the effect of the candidate policy can only be constrained to be between the lower and upper bound. For all problems studied in the paper, given the data and causal diagram, it is not theoretically possible to pinpoint a single value for the effect of the policy of interest.
>
> If the bound is wide, there is substantial uncertainty about the true value of the effect of the policy. In this case, the practitioner has to weigh this result along with other considerations (potential harm of the policy, cost of implementation, etc.) to arrive at a decision.

---

> > ### Author Response · Authors · 2024-08-13
> >
> > The discussion period is ending soon. We were wondering whether you had a chance to check our rebuttal. We hope to have answered all concerns to your satisfaction. If not, we would be happy to provide further comments and clarifications, let us know.
> >
> > Thank you again for your time and attention.

---

> > > ### Comment · Reviewer_LyZM · 2024-08-14
> > >
> > > I appreciate the responses to my questions and the numerics. I will increase my score.

---

### Author Rebuttal · Authors · 2024-08-06

we thank the reviewers for their time reviewing our work.

In this global rebuttal, we include an additional experiment that compares the bounds obtained according to the different propositions in Sec. 3 of the paper. This addresses specifically a comment from Reviewer LyZM. The details of the experiment are described in the attached PDF and are summarized below.

> Evaluation of bounds given by the different propositions introduced in the paper

Our simulations for this experiment are based on one of the data generating mechanisms used in the experimental section and described in more details in Example 4 Appendix C.1. The causal diagram is illustrated in the attached PDF and can also be found in the fourth row of Fig. 4 in the paper. We consider evaluating the policy in Eq. (16) of the paper and compute the bounds obtained by applying Prop. 2 (most conservative), Prop. 3 (using the partial adjustment set $W$ only), Prop. 4 (using the partial instrumental set $Z$ only), and finally Alg. 1 (that combines all propositions and is the proposed approach). The figure in the attached PDF gives the results over 10 seeds of the data and across multiple data sizes, highlighting the gain achieved by exploiting the causal structure using the proposed approaches. In this particular example, we are able to tighten the natural bounds -- approximately equal to [0.1, 0.9] -- to approximately [0.32, 0.77] with Alg. 1.

---

### Decision · Program_Chairs · 2024-09-25

**Decision:**

Accept (poster)

**Comment:**

I concur with the reviewers' positive view of the submission and think the paper would makes a nice addition to the nascent literature on this topic. I am happy to recommend its acceptance. However, I find that deferring all discussion of related work to the appendix to be inappropriate. Many papers on the precise subject of the paper (evaluation of policies/ATEs/HTEs under confounding) are cited but not discussed in the main text ([5,17,18,19,24,40]). I understand there are important differences to these works in the technical approach, and I appreciate that the works that the present papers' technical approach relies on are discussed somewhat in the main text, but additionally all works directly related to the problem itself, even with alternative technical approaches to it, should be discussed in the main text and compared to. So I ask the authors to prioritize discussing these in the main text, even over technical details or other extras that are more appropriate for deferring to the appendix. This should also ameliorate the concerns of the more negative reviewers regarding context in the literature. I fully trust that the authors can undertake this appropriately within the preparation of a camera ready version as this mostly involves moving content around from/to the appendix and/or adding a couple sentences here and there.